# FEDERATED CAUSAL INFERENCE ON MULTI-SITE OBSERVATIONAL DATA VIA PROPENSITY SCORE AGGREGATION

## ABSTRACT

Causal inference typically assumes centralized access to individual-level data. Yet, in practice, data are often decentralized across multiple sites, making centralization infeasible due to privacy, logistical, or legal constraints. We address this problem by estimating the Average Treatment Effect (ATE) from decentralized observational data via a Federated Learning (FL) approach, allowing inference through the exchange of aggregate statistics rather than individual-level data. We propose a novel method to estimate propensity scores by computing a federated weighted average of local scores with Membership Weights (MW)—probabilities of site membership conditional on covariates—which can be flexibly estimated using parametric or non-parametric classification models. Unlike density ratio weights (DW) from the transportability and generalization literature, which either rely on strong modeling assumptions or cannot be implemented in FL, MW can be estimated using standard FL algorithms and are more robust, as they support flexible, non-parametric models—making them the preferred choice in multi-site settings with strict data-sharing constraints. The resulting propensity scores are used to construct Federated Inverse Propensity Weighting (Fed-IPW) and Augmented IPW (Fed-AIPW) estimators. Unlike meta-analysis methods, which fail when any site violates positivity, our approach leverages heterogeneity in treatment assignment across sites to improve overlap. We show that Fed-IPW and Fed-AIPW perform well under site-level heterogeneity in sample sizes, treatment mechanisms, and covariate distributions. Both theoretical analysis and experiments on simulated and real-world data highlight their advantages over meta-analysis and related methods.

## 1 INTRODUCTION

The Average Treatment Effect (ATE) is a key causal estimand used to quantify the effect of a treatment on an outcome and is commonly employed as the primary measure of efficacy in evaluating new therapies, including vaccines, before regulatory approval (Polack et al., 2020). In Randomized Clinical Trials (RCTs), treatment assignment is randomized, ensuring that the observed association between treatment and outcome reflects a causal effect. Under this design, the ATE can be consistently estimated using a simple Difference-in-Means (DM) estimator (Splawa-Neyman, 1990), which can be further refined through covariate adjustment to reduce variance (FDA, 2023; EMA, 2024; Lei & Ding, 2021). However, RCTs are often expensive, time-consuming, or infeasible. In such cases, estimating treatment effects from observational data becomes the only viable alternative (Hernán, 2018; Hernán & Robins, 2006). Although such real-world data is abundant, drawing causal inferences from it is challenging due to confounding covariates, rendering the unadjusted DM estimator biased (Grimes & Schulz, 2002). Adjusting for confounders is thus essential (VanderWeele, 2019). This can be done by predicting counterfactual outcomes before averaging the differences (the G-formula plug-in estimator, Robins, 1986). Another approach is to weight individuals according to their treatment probability, emulating a randomized trial. For instance, the Inverse Propensity Weighting (IPW) estimator (Rosenbaum & Rubin, 1983) relies on estimating the *propensity score*—the probability of treatment given covariates. Doubly robust estimators such as the Augmented IPW (AIPW) (Bang & Robins, 2005) combine weighting with outcome modeling to remain consistent as long as either model is correctly specified.

Larger datasets improve the precision of treatment effect estimates, especially for underrepresented subgroups. Yet real-world data is typically decentralized—spread across hospitals, companies, or countries—making aggregation difficult, particularly in healthcare where privacy regulations, data ownership, and governance issues impede centralization. Federated Learning (FL) (Kairouz et al., 2021) offers a solution to train models across distributed data without sharing individual-level data. While FL has been largely applied to prediction tasks, its extension to causal inference remains limited. This problem is especially challenging in observational studies, where differences in covariate distributions and treatment assignment mechanisms across sites create multiple sources of heterogeneity that must be addressed without sharing raw data, while achieving results comparable to centralized analyses—an issue that remains largely unsolved.

**Contributions.** We propose federated (A)IPW estimators for decentralized observational data, moving beyond the aggregation of local ATE estimates used in meta-analysis (Riley et al., 2023). At the core of our approach is a flexible, primarily non-parametric strategy for federating propensity scores. Unlike prior methods that fit a single global parametric model via parameter averaging (Xiong et al., 2023) or federated gradient descent (Guo et al., 2025), we construct a global propensity score as a mixture of locally estimated models. This involves (1) local estimation of propensity scores at each site—accommodating heterogeneity in treatment assignment and flexibility in model choice—and (2) aggregation into a global score using Membership Weights (MW), i.e., the probability of site membership given covariates. MW can be estimated in a federated manner using flexible, potentially non-parametric classification models, ensuring both robustness and communication efficiency. In contrast, transferring Density Ratio Weights (DW) from the transportability and generalization literature to our setting requires strong modeling assumptions, as they are otherwise incompatible with FL constraints. Using our federated propensity scores, we build the Federated IPW (Fed-IPW) and its augmented variant (Fed-AIPW), derive their variances, and show they achieve equal or lower variance than meta-analysis estimators.

Our approach is particularly advantageous when overlap between treatment groups is poor or absent within sites. In such scenarios, cross-site collaboration becomes crucial, as combining sites increases overall overlap and enables treatment effect estimation that may be infeasible locally. Indeed, when treatment assignment mechanisms differ substantially—such as when one site treats a subgroup absent elsewhere—the combined dataset achieves markedly greater overlap, allowing treatment effects to be estimated that would otherwise be poorly identified in isolation. Additionally, our framework naturally accommodates heterogeneity in sample sizes, treatment policies, covariate distributions, and violations of positivity. Numerical experiments on simulated and real data confirm our theoretical findings and highlight the method's practical benefits.

**Related work.** Federated causal inference is still nascent. Khellaf et al. (2025) estimate federated G-formula ATE estimates across multiple RCTs by fitting parametric outcome models at each site via FL, achieving lower variance than DM. In observational settings, Vo et al. (2022) propose a federated Bayesian approach using Gaussian processes with a shared covariance kernel, but it requires sharing the first four moments of the data, limiting scalability and efficiency. Guo et al. (2025) learn a global propensity score via consensus voting over parametric parameters, retaining only sites meeting a shared specification. In contrast, our method assumes no common propensity score: each site may fit its own model.

To address site heterogeneity, Xiong et al. (2023) use a logistic propensity model with shared and site-specific parameters, federating only the common ones. Yin et al. (2025) fit a global model adjusting for covariates and site membership but limit heterogeneity to a site-specific scalar. By contrast, our method makes no structural assumptions, enabling fully nonparametric estimation with heterogeneous local models and relaxing the need for local overlap at each site.

A related body of work focuses on generalizing causal findings from multi-site source populations to a target population. Han et al. (2025) use density ratio weighting of local ATEs to adjust for covariate shift but assume homogeneous nuisance functions across sites and rely on meta-analysis of aggregate statistics. Guo et al. (2024) extend this idea by applying density ratio weights to aggregate local propensity scores to construct a target-specific score, which requires density estimation *within each treatment arm at each site*—demanding large sample sizes per arm for stable estimates. In both cases, non-parametric density ratio estimation is infeasible under FL constraints, as it requires sharing raw data or detailed covariate representations (e.g., kernel evaluations or high-dimensional histograms). In contrast, our MW-based approach leverages flexible parametric or non-parametric

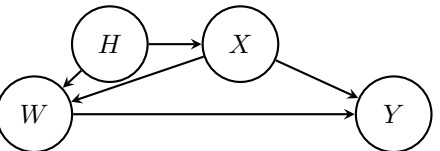

Figure 1: Graphical model for multi-site observational data.

supervised models (e.g., logistic regression, neural networks, gradient-boosted trees) for which federated training is already established, operating without sharing raw data, potentially at lower sample complexity, with full support for heterogeneous nuisance functions, and allowing the natural inclusion of external control arms.

## 2 PRELIMINARIES

### 2.1 ATE ESTIMATORS FROM CENTRALIZED MULTI-SITE OBSERVATIONAL DATA

In this section, we recall the key components of ATE estimation in a centralized multi-site setting. Following the potential outcomes framework (Rubin, 1974; Splawa-Neyman, 1990), we consider random variables $(X, H, W, Y(1), Y(0))$, where $X \in \mathbb{R}^d$ represents patient covariates, $H \in [K]$ indicates site membership, $W \in \{0, 1\}$ denotes the binary treatment, and $Y(1)$ and $Y(0)$ are the potential outcomes under treatment and control, respectively. We assume that the Stable Unit Treatment Values Assumption (SUTVA) holds, so that the observed outcome is $Y = WY(1) + (1 - W)Y(0)$, and that the potential outcomes are uniformly bounded. In the centralized setting, we observe $n = \sum_{k=1}^{K} n_k$ observations of independently and identically distributed (i.i.d.) tuples $(H_i, X_i, W_i, Y_i)_{i \in 1,\dots,n} \sim \mathcal{P}^{\otimes n}$, with $n_k = \sum_{i=1}^{n} \mathbb{1}_{\{H_i=k\}}$ the number of observations in site $k$. We aim to estimate the ATE defined as the risk difference $\tau = \mathbb{E}[Y(1) - Y(0)] = \mathbb{E}[\mathbb{E}[Y(1) - Y(0) \mid H]]$, where the expectation is taken over the population $\mathcal{P}$. To be able to identify the ATE, we assume unconfoundedness (standard in causal inference) and further consider Assumption 2, which is specific to the multi-site setting.

**Assumption 1** (Unconfoundedness). $\{Y(0), Y(1)\} \perp\!\!\!\perp W \mid (X, H)$.

**Assumption 2** (Ignorability on sites). $\{Y(0), Y(1)\} \perp\!\!\!\perp H \mid X$.

Assumptions 1 and 2 imply $\{Y(0), Y(1)\} \perp H \mid (X, W)$ (Dawid, 1979, Lemma 4.3). This entails the *no center-outcome association* condition (Robertson et al., 2021), yielding the testable null hypothesis $\mathbb{E}[Y \mid X, W, H] = \mathbb{E}[Y \mid X, W]$. This ensures that $X$ forms a sufficient set of covariates for confounding adjustment. Our setting is depicted in the graphical model in Figure 1, highlighting that we remove any direct effect of the site on the outcome.

*Remark* (Accommodating site effects). Our approach can be extended to handle site effects when site-level covariates are available, providing a weaker and testable alternative to Assumption 2. This extension, along with the associated experiments, is discussed in Appendix F.

We define $\mu_w(x) = \mathbb{E}[Y \mid X = x, W = w]$ for $w \in \{0, 1\}$, and let $\tau(x) = \mu_1(x) - \mu_0(x) = \mathbb{E}[Y(1) - Y(0) \mid X = x]$ be the Conditional Average Treatment Effect (CATE). The oracle propensity score is denoted by $e(x) = \mathbb{P}(W \mid X = x)$, and we consider the weak (global) overlap assumption (Wager, 2024).

**Assumption 3** (Global overlap). $\mathcal{O}_{\text{global}} = \mathbb{E}[(e(X)(1 - e(X)))^{-1}] < +\infty$.

Assumption 3 is crucial for propensity score-based estimators, as it states that every region of the covariate space has a non-zero probability of receiving both treatments. A lower value of $\mathcal{O}_{\text{global}}$ indicates that these probabilities lie further away from 0 and 1, which corresponds to better overlap. For further insights on overlap, see Li et al. (2018a;b), and for a "misoverlap" metric, refer to Clivio et al. (2024).

With Assumptions 1, 2 and 3, the ATE is identifiable as $\tau = \mathbb{E}\left[\frac{WY}{e(X)} - \frac{(1-W)Y}{1-e(X)}\right]$ (see Appendix A.1). Throughout the paper, we denote oracle ATE estimators, which assume knowledge of the nuisance

components $e, \mu_0, \mu_1$, by a superscript $*$. We define the *Oracle multi-site centralized estimators* as follows:

$$\hat{\tau}_{\text{IPW}}^* = \frac{1}{n} \sum_{i=1}^n \tau_{\text{IPW}}(X_i; e), \qquad \hat{\tau}_{\text{AIPW}}^* = \frac{1}{n} \sum_{i=1}^n \tau_{\text{AIPW}}(X_i; e, \mu_1, \mu_0), \tag{1}$$

where $\tau_{\text{IPW}}(X_i; e) = \frac{W_i Y_i}{e(X_i)} - \frac{(1-W_i)Y_i}{1-e(X_i)}$ and $\tau_{\text{AIPW}}(X_i; e, \mu_1, \mu_0) = \mu_1(X_i) - \mu_0(X_i) + \frac{W_i(Y_i - \mu_1(X_i))}{e(X_i)} - \frac{(1-W_i)(Y_i - \mu_0(X_i))}{1-e(X_i)}$. These oracle estimators are unbiased and asymptotically normal.

**Theorem 1.** *Under Assumptions 1, 2 and 3, we have $\sqrt{n}(\hat{\tau} - \tau) \to \mathcal{N}(0, V)$ with*

$$\begin{cases} V_{\text{IPW}} = \mathbb{E}\left[\frac{Y(1)^2}{e(X)}\right] + \mathbb{E}\left[\frac{Y(0)^2}{1-e(X)}\right] - \tau^2, \\ V_{\text{AIPW}} = \mathbb{V}\left[\tau(X)\right] + \mathbb{E}\left[\left(\frac{(Y-\mu_1(X))^2}{e(X)}\right)\right] + \mathbb{E}\left[\left(\frac{(Y-\mu_0(X))^2}{1-e(X)}\right)^2\right]. \end{cases}$$

The above asymptotic variances align with those in the single-site setting (Hirano et al., 2003), as detailed in Appendix A.2. However, in practice, the propensity score and outcome models are typically unknown and must be estimated from data. This creates a challenge in the decentralized setting, where centralizing data to compute $\mu_1$, $\mu_0$, and $e$ is not feasible. Therefore, the estimators in Definition 1 need to be adapted to this setting. Importantly, the (non-oracle) AIPW estimator is inherently *doubly robust*, remaining consistent as long as either the outcome or the propensity score model is correctly specified (Chernozhukov et al., 2018).

## 2.2 META-ANALYSIS ESTIMATORS

We now turn to a decentralized setting in which the $K$ sites cannot share individual-level data. A natural baseline for estimating the ATE across sites is a two-stage meta-analysis approach (Burke et al., 2017), wherein each site independently estimates the relevant nuisance parameters and communicates only the resulting ATE estimates for aggregation. In this setting, we need the following assumption.

**Assumption 4** (Local overlap). $\forall k \in [K], \mathcal{O}_k = \mathbb{E}\left[(e(X)(1-e(X)))^{-1} \mid H = k\right] < +\infty$.

Assumption 4 is much stronger than global overlap (Assumption 3), as it must hold at every site. Denoting by $e_k(x) = \mathbb{P}(W = 1 \mid X = x, H = k)$ the oracle local propensity score at site $k$, we can define the oracle meta-analysis estimators as follows:

$$\hat{\tau}_{\text{IPW}}^{\text{meta}^*} = \sum_{k=1}^K \frac{n_k}{n} \hat{\tau}_{\text{IPW}}^{(k)}, \qquad \hat{\tau}_{\text{AIPW}}^{\text{meta}^*} = \sum_{k=1}^K \frac{n_k}{n} \hat{\tau}_{\text{AIPW}}^{(k)}, \tag{2}$$

where $\hat{\tau}_{\text{IPW}}^{(k)} = \frac{1}{n_k} \sum_{i=1}^{n_k} \tau_{\text{IPW}}(X_i; e_k)$ and $\hat{\tau}_{\text{AIPW}}^{(k)} = \frac{1}{n_k} \sum_{i=1}^{n_k} \tau_{\text{AIPW}}(X_i; e_k, \mu_1, \mu_0)$ are the local estimators at site $k$. While alternative aggregation weights—such as the inverse variance of local estimates—can be considered, they produce, in our setting, biased estimates of the global ATE $\tau = \sum_{k=1}^K \rho_k \tau_k$, where $\rho_k = \mathbb{P}(H = k)$ and $\tau_k = \mathbb{E}[Y(1) - Y(0) \mid H = k]$ is the local ATE. This bias appears whenever the $\tau_k$ differ, which commonly occurs when covariate distributions vary across sites and treatment effects are heterogeneous (i.e., depend on covariates), see (Berenfeld et al., 2025).

**Theorem 2.** *Under Assumptions 1, 2 and 4, the oracle meta-analysis estimators are unbiased for the ATE with asymptotic variances*

$$V_{\text{IPW}}^{\text{meta}^*} = \sum_{k=1}^K \rho_k V_{\text{IPW}}^{(k)} + \mathbb{V}\left[\tau_H\right], \qquad V_{\text{AIPW}}^{\text{meta}^*} = \sum_{k=1}^K \rho_k V_{\text{AIPW}}^{(k)} + \mathbb{V}\left[\tau_H\right],$$

*with within-site variance*

$$\begin{cases} V_{\text{IPW}}^{(k)} = \mathbb{E}\left[\frac{Y(1)^2}{e_k(X)} \mid H = k\right] + \mathbb{E}\left[\frac{Y(0)^2}{1-e_k(X)} \mid H = k\right] - \tau_k^2 \\ V_{\text{AIPW}}^{(k)} = \mathbb{V}\left[\tau(X) \mid H = k\right] + \mathbb{E}\left[\left(\frac{(Y-\mu_1(X))^2}{e_k(X)}\right)^2 \mid H = k\right] + \mathbb{E}\left[\left(\frac{(Y-\mu_0(X))^2}{1-e_k(X)}\right) \mid H = k\right], \end{cases}$$

*and $\mathbb{V}\left[\tau_H\right] = \mathbb{V}\left[\mathbb{E}\left[Y(1) - Y(0) \mid H\right]\right]$ the between-sites variance of the local ATEs.*

This result is proved in Appendix A.3. A key limitation of meta-analysis estimators is their reliance on Assumption 4, which is fragile and often violated—for instance, when a site applies a deterministic treatment policy (treating all patients or only a subgroup). In such cases, these estimators are ill-defined, yielding biased ATE estimates. To address this, we propose a federated approach that constructs the global propensity score $e$ as a weighted combination of local scores $e_k$, enabling valid inference even without local overlap.

## 3 FEDERATED ESTIMATORS VIA PROPENSITY SCORE AGGREGATION

### 3.1 ORACLE FEDERATED ESTIMATORS

As discussed before, existing federated causal inference methods often rely on restrictive assumptions—such as a common propensity score across sites (Guo et al., 2025), site differences limited to intercept shifts (Yin et al., 2025), or predefined shared structures (Xiong et al., 2023). In practice, treatment assignment frequently varies across sites due to differences in norms, resources, or clinical practices. This heterogeneity implies that the global propensity score takes the form of a *weighted combination* of the site-specific scores, that is $e(X) = \sum_{k=1}^{K} \omega_k(X) e_k(X)$, see Appendix A.4. These weights $\{\omega_k\}_{k \in [K]}$ can be written as density ratio weights (DW):

$$e(x) = \sum_{k=1}^{K} \underbrace{\rho_k \frac{f_k(x)}{f(x)}}_{=\omega_k(x)} e_k(x), \tag{3}$$

where $f_k$ and $f$ are the covariate densities locally and globally. Similar weights are used in transportability methods that reweight data to match a target population (Han et al., 2023; 2025; Guo et al., 2024), though here the objective is to recover the global propensity score across the super-population defined by the $K$ participating sites. Unfortunately, estimating DW $\omega_k$ in a federated setting requires modeling the $f_k$'s, which entails strong distributional assumptions and becomes challenging in high dimensions.

Instead, we propose to rewrite the $\omega_k$'s as *Membership Weights (MW)*:

$$e(x) = \sum_{k=1}^{K} \underbrace{\mathbb{P}(H = k \mid X = x)}_{= \omega_k(x)} e_k(x), \tag{4}$$

which represent the probability of site membership given the covariates. Unlike DW, MW do not require explicit density modeling and can be estimated directly via federated parametric (e.g., logistic regression, neural networks) or non-parametric (e.g., gradient-boosted trees) classification models, providing a flexible, communication-efficient alternative and making it the preferred choice for federated settings. We refer to Section 3.2 for more details on the federated estimation of MW and its advantages over DW.

Equations 3 and 4 enable combining locally estimated propensity scores $e_k$ into a global propensity score using globally learned weights $\omega_k(x)$. Building on this decomposition, we define our oracle Federated IPW and AIPW estimators (Fed-(A)IPW). We define the *Oracle federated estimators* as follows:

$$\hat{\tau}_{\text{IPW}}^{\text{fed}^*} = \sum_{k=1}^{K} \frac{n_k}{n} \hat{\tau}_{\text{IPW}}^{\text{fed}(k)}, \qquad \hat{\tau}_{\text{AIPW}}^{\text{fed}^*} = \sum_{k=1}^{K} \frac{n_k}{n} \hat{\tau}_{\text{AIPW}}^{\text{fed}(k)}, \tag{5}$$

where $\hat{\tau}_{\text{IPW}}^{\text{fed}(k)} = \frac{1}{n_k} \sum_{i=1}^{n_k} \tau_{\text{IPW}}(X_i; e)$ and $\hat{\tau}_{\text{AIPW}}^{\text{fed}(k)} = \frac{1}{n_k} \sum_{i=1}^{n_k} \tau_{\text{AIPW}}(X_i; e, \mu_1, \mu_0)$ rely on the global propensity score $e(x) = \sum_{k=1}^{K} \omega_k(X) e_k(X)$.

Theorem 3 (proved in Appendix A.5) establishes that, in the oracle setting, Fed-(A)IPW estimators attain the same efficiency as their centralized counterparts.

**Theorem 3.** *Under Assumptions 1, 2, and 3, the oracle federated estimators (Equation 5) are identical to the oracle centralized estimators (Equation 1).*

Theorem 4 (proved in Appendix A.6) further shows that even when local overlap (Assumption 4) holds, federated estimators have lower variance than meta-analysis estimators.

**Theorem 4.** *Under Assumptions [1], [2] and [4], we have:*

$$\mathbb{V}[\hat{\tau}_{\mathrm{IPW}}^*] = \mathbb{V}[\hat{\tau}_{\mathrm{IPW}}^{\mathrm{fed}^*}] \leq \mathbb{V}[\hat{\tau}_{\mathrm{IPW}}^{\mathrm{meta}^*}], \qquad \mathbb{V}[\hat{\tau}_{\mathrm{AIPW}}^*] = \mathbb{V}[\hat{\tau}_{\mathrm{AIPW}}^{\mathrm{fed}^*}] \leq \mathbb{V}[\hat{\tau}_{\mathrm{AIPW}}^{\mathrm{meta}^*}],$$

*with equality when the local propensity scores are identical across sites.*

This variance reduction arises for two reasons. First, decomposing $e$ as a weighted sum of $e_k$'s marginalizes over $H \mid X$, eliminating unnecessary adjustment for site membership and thereby reducing variance. Second, our federated approach improves overlap compared with meta-analysis, as formalized below.

**Theorem 5** (Overlap improvement). $0 \leq \mathcal{O}_{\mathrm{global}} \leq \sum_{k=1}^K \rho_k \mathcal{O}_k$.

Theorem [5] (proved in Appendix [A.7]) shows that global overlap is always at least as good as the worst local overlap. Even when local overlap holds, sites with poor overlap benefit from the federated approach because the global score $e(x)$ is more bounded away from 0 and 1 than the local scores $\{e_k(x)\}_{k \in [K]}$. Notably, sites with poor individual overlap can even improve the overall overlap of the federated population, as illustrated in the following example.

*Example.* Let $K = 2$ with $X_i = 1$ in both sites, $\mathbb{P}(H_i \mid X_i) = 0.5$, $e_1(X_i) = 0.99$ and $e_2(X_i) = 0.01$, leading to $e(X_i) = \sum_{k=1}^2 0.5 \times e_k(X_i) = 0.5$. Local overlaps are poor, $\mathcal{O}_1 = \mathcal{O}_2 = (0.99 \times 0.01)^{-1} \approx 101$, whereas the global overlap is $\mathcal{O}_{\mathrm{global}} = (0.5 \times 0.5)^{-1} = 4$—the optimal value achieved in a randomized trial with $50\%$ treatment probability. This illustrates how heterogeneity in treatment assignments can enhance global overlap and enable more robust causal inference.

## 3.2 FEDERATED ESTIMATION

We now move beyond oracle estimators and describe how to implement our Fed-(A)IPW estimators in a practical federated learning setting. Constructing the global score propensity score requires two steps, which can be executed in parallel: each site $k$ estimates and shares a local propensity score $\hat{e}_k(x)$; and the sites collaboratively estimate federated weights $\{\hat{\omega}_k(x)\}_{k \in [K]}$. Fed-AIPW adds a third step to train outcome models $\hat{\mu}_0, \hat{\mu}_1$ via federated learning. We detail how to estimate $\{\hat{e}_k(x), \hat{\omega}_k(x)\}_k$ and $\hat{\mu}_0, \hat{\mu}_1$ below.

**Local propensity scores.** Each $e_k$ can be estimated using any probabilistic binary classifier, either parametric (e.g., logistic regression or neural networks) or non-parametric ( (e.g., generalized random forests, Lee et al., 2010). A key advantage of our approach is flexibility: sites can use different estimation methods tailored to local data or computational constraints. It also does *not* require Assumption [4]: local scores may approach 0 or 1 provided the *global* score remains bounded away from these extremes.This, in particular, enables the integration of external control arms (FDA, 2023; EMA, 2023), where some sites have $e_k(X) = 0$ for all control patients yet still contribute to the global analysis.

**Federated weights: density ratio vs. membership.** Parametric density ratio weights $\hat{\omega}_k^{\mathrm{DW}}(x) = \rho_k \frac{\hat{f}_k(x)}{\hat{f}(x)}$ can be implemented in a one-shot fashion by sharing local density parameters with the server, which then reconstructs the global mixture and computes the weights. A common choice is to assume parametric covariate distributions (say, Gaussian), estimate $(\hat{\mu}_k, \hat{\Sigma}_k)$ locally, and transmit them once to the server. This requires to communicate $O(Kd^2)$ parameters and is highly sensitive to model misspecification—an issue that becomes critical in high dimensions. Nonparametric density estimation would relax these assumptions but is statistically inefficient and does not yet have practical federated implementations.

In contrast, our membership weights $\hat{\omega}_k^{\mathrm{MW}}(x) = \widehat{\mathbb{P}}(H = k \mid X = x)$ can be learned with any probabilistic multiclass classifier trained federatively—for example, logistic regression for simplicity and interpretability, or neural networks to capture complex nonlinearities. Such models are readily supported by modern FL algorithms and software libraries. Using the standard FedAvg algorithm (McMahan et al., 2017) requires exchanging $TKP$ floats (training rounds × sites × model parameters), which is feasible for models of practical size and a large number of sites. Non-parametric classifiers like random forests (Hauschild et al., 2022) and gradient-boosted trees (Li et al., 2020) have also been adapted to the federated setting.

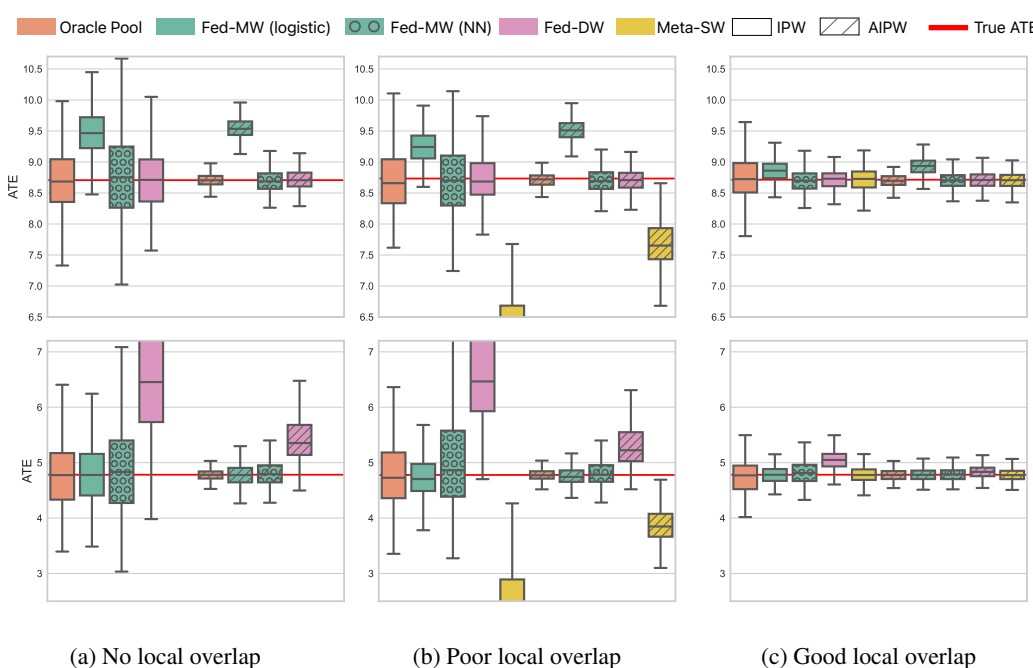

(a) No local overlap      (b) Poor local overlap      (c) Good local overlap

Figure 2: Synthetic data: DGP A (top) and DGP B (bottom).

**Outcome models.** To construct the doubly robust Fed-AIPW estimator, $\mu_0, \mu_1$ are trained federatively, as in Khellaf et al. (2025). As for MW, standard FL algorithms and librairies can be used to train a wide range of parametric and non-parametric supervised learning models.

*Remark* (Missing values). Our approach can naturally accommodate missing data. Local propensity scores can be estimated consistently with logistic regression or random forests (Jiang et al., 2020; Josse et al., 2024). For membership weights, constant imputation combined with federated random forests provides a consistent solution (Le Morvan et al., 2021), whereas density ratio weights would require adapting more complex federated EM algorithms (Dieuleveut et al., 2021; Marfoq et al., 2021). Finally, doubly robust estimators with missing data can be obtained via non-parametric federated outcome models (Mayer et al., 2020).

## 4 EXPERIMENTS

**Synthetic data.** We consider $K = 3$ sites and $d = 10$ covariates. Two data-generating processes (DGPs) are used. In *DGP A*, each site $k$ independently samples $n_k = 650$ individuals from a site-specific multivariate Gaussian distribution $\mathcal{N}(\mu_k, \Sigma_k)$. In *DGP B*, a total of $n = 4000$ individuals are first drawn from a bimodal Gaussian mixture and then assigned to sites according to a multinomial logistic model based on their covariates. We vary within-site overlap to mimic different practical scenarios: *No local overlap* ($\mathcal{O}_2 = +\infty$, the second site has no treated individuals), *Poor local overlap* ($\mathcal{O}_2 \approx 10^7$), and *Good local overlap* ($\mathcal{O}_2 \approx 4.6$). The outcome models $\mu_1, \mu_0$ are shared across sites and specified as polynomial functions with interactions. For comparison, we also generate data consistent with the setting in Xiong et al. (2023) (Figure 3). All results are averaged over 800 simulation runs; full details are provided in Appendix C.

We evaluate our proposed **Fed-IPW** and **Fed-AIPW** using the **MW** weights estimated either via federated multinomial logistic regression (well specified in *DGP B*) or via a two-layer Neural Network (NN). We compare these methods against several competitors: **Fed-IPW** and **Fed-AIPW** with the alternative **DW** weights based on Gaussian density estimation (well specified in *DGP A*); the **Centralized Oracle** (Def. 1); meta-analysis IPW/AIPW with sample-size weighting (**Meta-SW**) (Def. 2); and the one-shot inverse-variance weighted IPW estimator (**1S-IVW**) of Xiong et al. (2023), evaluated under favorable conditions with shared propensity-score parameters across sites.

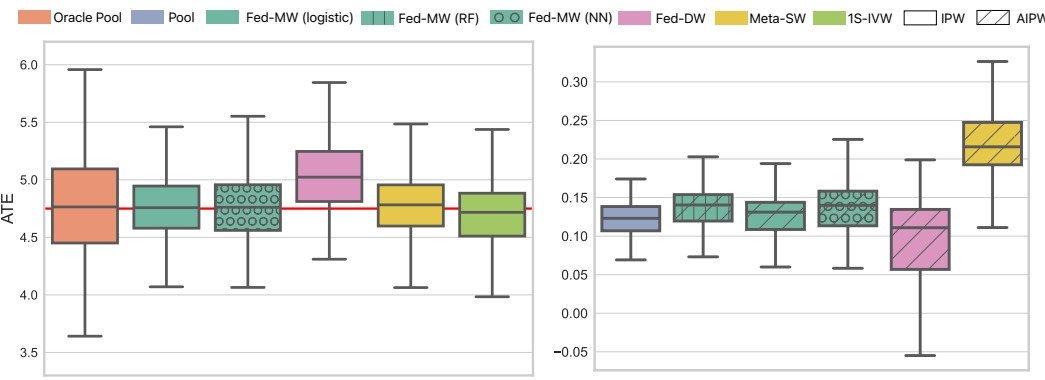

Figure 3: Comparison to Xiong et al. (2023) (IPW).  Figure 4: AIPW estimates on Traumabase.

For all estimators, propensity scores are fit via logistic regression and outcome models via linear regression, implying misspecification in the latter.

Figures 2a–2c (top: *DGP A*; bottom: *DGP B*) summarize the results under the three overlap regimes. Before discussing each setting, we highlight several general observations. First, Fed-IPW-DW is unbiased only under *DGP A*, where the Gaussian specification holds, and exhibits bias under *DGP B*. By contrast, Fed-IPW-MW adapts across data-generating processes: with logistic membership weights it is unbiased in *DGP B*, and with a more flexible neural network classifier it is unbiased for both *DGP A* and *DGP A*, reflecting the greater modeling flexibility of MW relative to DW's restrictive density specification. Second, Fed-AIPW enjoys its doubly robust property and remains unbiased across all overlap levels by double robustness; despite misspecified linear outcome models, accurate propensity estimation ensures consistency. Finally, Fed-IPW typically attains lower variance than the centralized oracle IPW, consistent with well-documented efficiency gains from using estimated propensity scores (Hirano et al., 2003).

In the *No local overlap* setting (Figure 2a), meta-analysis estimators are undefined because one site has no treated individuals. Our federated estimators remain unbiased under both DGPs, as Assumption 3 holds (global overlap $\mathcal{O}_{\mathrm{global}} \approx 6.22$). In the *Poor local overlap* setting (Figure 2b), site 2 exhibits weak overlap, leading to bias and instability in meta-analysis estimators, including Meta-AIPW, as both the propensity scores and local outcome models are inaccurate. This issue is mitigated in the global dataset (see Figure 5 in the Appendix), allowing our federated estimators to remain reliable. In the *Good local overlap* setting (Figure 2c), all methods are unbiased, but federated estimators achieve the smallest variance.

Figure 3 considers a setting where all local propensity scores share a common subset of 5 of 10 logistic regression coefficients with data generated from *DGP B*. This setup matches the assumptions of the 1S-IVW method of Xiong et al. (2023), which relies on prior knowledge of the shared parameters to aggregate them—an assumption not required by our method. Fed-MW remains unbiased and attains the lowest variance, matching that of 1S-IVW, even without access to the additional information about shared parameters.

Additional simulations provided in Appendix E examine scenarios with more sites (Table 4), non-parametric estimation (Table 5), and local propensity model misspecification (Table 6), further confirming the robustness of our approach. In Appendix F.2, we demonstrate the adaptability of our framework to site effects by leveraging site-level covariates, showing superior performance over meta-analysis when local sample sizes are small.

**Real data.** We analyze the multi-site Traumabase cohort (Mayer et al., 2020; Colnet et al., 2024) to estimate the effect of tranexamic acid on mortality across $K = 14$ centers. The local datasets are highly imbalanced in site sizes (106 to 2,092 patients) and treatment arms (e.g., site 11: 4 treated vs. 121 controls); there are 17 covariates and $n = 8,248$ patients in total, of whom 638 were treated. Covariates are standardized federatively by sharing site means and variances. We focus on AIPW estimators: local propensities $e_k$ are estimated using the R package `grf`'s `probability_forest` function (Tibshirani et al., 2018) on large sites, and logistic regressions on smaller ones; outcome

models $\mu_1, \mu_0$ are trained with FedAvg logistic regression (5,000 rounds, 1 local epoch, step size $\eta = 0.1$); MW are learned with a FedAvg neural network (NN, one hidden layer, 128 units) and a federated logistic regression. We also report a centralized random forest (RF) MW only as a benchmark—we do not develop a federated RF in this work. DW are based on Gaussian density estimation. Competitors include Meta-SW (same local models, computed only on sites with enough treated units—where the number of treated observations exceeds the covariate dimension) and a centralized AIPW benchmark obtained using `probability_forest` function on the pooled data. Empirical confidence intervals are constructed from 150 bootstrap resamples.

Figure 4 shows that our federated MW estimators closely match the centralized benchmark and exhibit lower variance than Meta-SW, which in this application departs from the centralized estimate. Among federated methods, Fed-MW are closest to the nonparametric centralized estimators—reflecting their more flexible modeling of propensities and membership—whereas Fed-DW is less stable, likely due to noisy per-site covariance estimates for a 17-dimensional Gaussian ($\approx 17^2$ parameters) in small sites.

## 5 CONCLUSION, LIMITATIONS AND FUTURE WORK

We propose a theoretically grounded framework for federated causal inference that leverages membership weights to construct valid pseudo-populations across silos. These weights, estimated via flexible parametric or nonparametric models, improve overlap and yield more stable ATE estimates, without sharing raw data. Our framework accommodates heterogeneous local propensity score estimation strategies, supports external control arms, and remains robust to even extreme local treatment–control imbalances. Although sufficiently large per-site datasets are still needed—particularly in high-dimensional settings—our approach is especially well suited to a moderate number of large silos, where FL most effectively increases effective sample size.

Promising avenues for future work include principled handling of site effects (i.e., relaxing Assumption 2) and extending our framework to CATE estimation. While the ATE remains central in econometrics, biomedicine, and public policy, considering CATE is important for moving towards personalization. Our work lays a foundation for federated CATE estimation: most learners (T-, S-, X-, and R-learners) rely on nuisance quantities such as propensity scores, and our federated estimation framework could be incorporated into these methods, although further work is needed to assess its formal properties and practical performance.

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

# A PROOFS

## A.1 ATE IDENTIFICATION AND SUFFICIENT CONDITIONS

*Proof.* Under Assumptions 1 and 2, the variables in $X$ form a sufficient adjustment set. We identify the ATE as follows:

$$
\begin{aligned}
\tau &= \mathbb{E}\left[Y(1) - Y(0)\right] \\
&= \mathbb{E}\left[\mathbb{E}\left[Y(1) \mid X, H\right] - \mathbb{E}\left[Y(0) \mid X, H\right]\right] && \text{Law of Iterated Expectations} \\
&= \mathbb{E}\left[\mathbb{E}\left[Y(1) \mid X, H, W = 1\right] - \mathbb{E}\left[Y(0) \mid X, H, W = 0\right]\right] && \text{Assumption 1} \\
&= \mathbb{E}\left[\mathbb{E}\left[Y(1) \mid X, W = 1\right] - \mathbb{E}\left[Y(0) \mid X, W = 0\right]\right] && \text{Assumption 2 and Lemma 4.3 Dawid (1979)} \\
&= \mathbb{E}\left[\mathbb{E}\left[Y \mid X, W = 1\right] - \mathbb{E}\left[Y \mid X, W = 0\right]\right] && \text{Consistency} \\
&= \mathbb{E}\left[\frac{WY}{e(X)} - \frac{(1-W)Y}{1-e(X)}\right] && \text{Assumption 3}
\end{aligned}
$$

$\square$

## A.2 PROOF OF THEOREM 1

*Proof.* **Unbiasedness.** We show the unbiasedness of the first term of the IPW estimator. The derivation for the second term is symmetric.

$$
\begin{aligned}
\mathbb{E}\left[\frac{WY}{e(X)}\right] &= \mathbb{E}\left[\mathbb{E}\left[\frac{WY(1)}{e(X)}\bigg| X, H\right]\right] && \text{SUTVA} \\
&= \mathbb{E}\left[\frac{\mathbb{E}\left[W \mid X, H\right]\mathbb{E}\left[Y(1) \mid X, H\right]}{e(X)}\right] && \text{Assumption 1} \\
&= \mathbb{E}\left[\frac{e_H(X)\mathbb{E}\left[Y(1) \mid X, H\right]}{e(X)}\right] && \text{Definition of local score } e_H(X) \\
&= \mathbb{E}\left[\frac{e_H(X)\mathbb{E}\left[Y(1) \mid X\right]}{e(X)}\right] && \text{Assumption 2} \\
&= \mathbb{E}_X\left[\frac{\mathbb{E}\left[Y(1) \mid X\right]}{e(X)}\mathbb{E}_H\left[e_H(X) \mid X\right]\right] && \text{Iterated Expectation over } H \\
&= \mathbb{E}_X\left[\frac{\mathbb{E}\left[Y(1) \mid X\right]}{e(X)}\sum_{k=1}^{K}\mathbb{P}(H = k \mid X)e_k(X)\right] \\
&= \mathbb{E}_X\left[\frac{\mathbb{E}\left[Y(1) \mid X\right]}{e(X)}e(X)\right] && \text{Definition of global } e(X) \\
&= \mathbb{E}\left[\mathbb{E}[Y(1) \mid X]\right] = \mathbb{E}[Y(1)]
\end{aligned}
$$

Note that the step applying Assumption 2 relies on the consequence that $\mathbb{E}[Y(1) \mid X, H] = \mathbb{E}[Y(1) \mid X]$ (which is derived from Assumption 2 combined with Assumption 1).

Similarly, we have $\mathbb{E}\left[\frac{(1-W)Y}{1-e(X)}\right] = \mathbb{E}[Y(0)]$, so that $\mathbb{E}\left[\hat{\tau}^{\text{IPW}^*}\right] = \tau$, which concludes the proof of unbiasedness of the oracle multi-site centralized IPW estimator.

For the AIPW estimator, we note that $\mathbb{E}[W(Y - \mu_1(X))/e(X)] = 0$ following the same logic as above (replacing $Y(1)$ with the residual $Y(1) - \mu_1(X)$ which has mean zero given $X$). Thus:

$$
\begin{aligned}
\mathbb{E}\left[\hat{\tau}^{\text{AIPW}^*}\right] &= \mathbb{E}\left[\mu_1(X) - \mu_0(X) + \frac{W(Y - \mu_1(X))}{e(X)} - \frac{(1-W)(Y - \mu_0(X))}{1-e(X)}\right] \\
&= \mathbb{E}[\mu_1(X) - \mu_0(X)] + 0 - 0 \\
&= \mathbb{E}[\tau(X)] = \tau
\end{aligned}
$$

**Variance.** As we consider uniformly bounded potential outcomes, that is $\forall w \in \{0, 1\}, \exists (L, U) \in \mathbb{R}^2, L < Y(w) < U$, and Assumption 3, we have that $\mathbb{E}\left[\frac{Y_i(1)^2}{e(X_i)}\right] < \infty$ and $\mathbb{E}\left[\frac{Y_i(0)^2}{1-e(X_i)}\right] < \infty$, so the quantities that follow are well defined:

$$
\begin{aligned}
\mathbb{V}\left[\hat{\tau}^{\mathrm{IPW}^*}\right] &= \mathbb{V}\left[\frac{1}{n}\sum_{i=1}^{n}\left(\frac{W_i Y_i}{e(X_i)} - \frac{(1-W_i)Y_i}{1-e(X_i)}\right)\right] \\
&= \frac{1}{n^2}\sum_{i=1}^{n}\mathbb{V}\left[\frac{W_i Y_i}{e(X_i)} - \frac{(1-W_i)Y_i}{1-e(X_i)}\right] && \text{i.i.d. observations} \\
&= \frac{1}{n^2}\sum_{i=1}^{n}\left(\mathbb{E}\left[\left(\frac{W_i Y_i}{e(X_i)} - \frac{(1-W_i)Y_i}{1-e(X_i)}\right)^2\right] - \tau^2\right) && \text{unbiasedness} \\
&= \frac{1}{n^2}\sum_{i=1}^{n}\left(\mathbb{E}\left[\left(\frac{W_i Y_i}{e(X_i)}\right)^2\right] + \mathbb{E}\left[\left(\frac{(1-W_i)Y_i}{1-e(X_i)}\right)^2\right] - 2\mathbb{E}\left[0\right] - \tau^2\right) \\
&= \frac{1}{n^2}\sum_{i=1}^{n}\left(\mathbb{E}\left[\left(\frac{W_i Y_i}{e(X_i)}\right)^2\right] + \mathbb{E}\left[\left(\frac{(1-W_i)Y_i}{1-e(X_i)}\right)^2\right] - \tau^2\right)
\end{aligned}
$$

$$
\begin{aligned}
\mathbb{E}\left[\left(\frac{W_i Y_i}{e(X_i)}\right)^2\right] &= \mathbb{E}\left[\mathbb{E}\left[\left(\frac{W_i Y_i}{e(X_i)} \mid X_i\right)^2\right]\right] \\
&= \mathbb{E}\left[\frac{\mathbb{E}\left[(W_i Y_i \mid X_i)^2\right]}{e(X_i)^2}\right] \\
&= \mathbb{E}\left[\frac{\mathbb{E}\left[W_i \mid X_i\right]\mathbb{E}\left[Y_i^2 \mid X_i, W_i = 1\right]}{e(X_i)^2}\right] && \text{Assumption 1} \\
&= \mathbb{E}\left[\frac{\mathbb{E}\left[Y_i(1)^2 \mid X_i\right]}{e(X_i)}\right] && \text{SUTVA} \\
&= \mathbb{E}\left[\frac{Y_i(1)^2}{e(X_i)}\right]
\end{aligned}
$$

Similarly, $\mathbb{E}\left[\left(\frac{(1-W_i)Y_i}{1-e(X_i)}\right)^2\right] = \mathbb{E}\left[\frac{Y_i(0)^2}{1-e(X_i)}\right]$. Then, the variance of the oracle multi-site IPW estimator is

$$
\mathbb{V}\left[\hat{\tau}^{\mathrm{IPW}^*}\right] = \frac{1}{n}\left(\mathbb{E}\left[\frac{Y_i(1)^2}{e(X_i)}\right] + \mathbb{E}\left[\frac{Y_i(0)^2}{1-e(X_i)}\right] - \tau^2\right).
$$

For the variance of the oracle multi-site centralized AIPW, we first notice that since $\mathbb{E}\left[Y_i(w) - \mu_w(X_i) \mid X_i\right] = 0$ for treatment $w$, we have

$$
\begin{aligned}
A &= \mathrm{Cov}\left(\tau(X_i), \frac{W_i(Y_i - \mu_1(X_i))}{e(X_i)} - \frac{(1-W_i)(Y_i - \mu_0(X_i))}{1-e(X_i)}\right) \\
&= \mathbb{E}\left[\tau(X_i)\left(\frac{W_i(Y_i - \mu_1(X_i))}{e(X_i)} - \frac{(1-W_i)(Y_i - \mu_0(X_i))}{1-e(X_i)}\right)\right] \\
&= \mathbb{E}\left[\tau(X_i)\mathbb{E}\left[\frac{W_i(Y_i - \mu_1(X_i))}{e(X_i)} \mid X_i\right]\right] - \mathbb{E}\left[\tau(X_i)\mathbb{E}\left[\frac{(1-W_i)(Y_i - \mu_0(X_i))}{1-e(X_i)} \mid X_i\right]\right] \\
&= \mathbb{E}\left[\tau(X_i)\frac{e(X_i)\mathbb{E}\left[Y_i(1) - \mu_1(X_i) \mid X_i\right]}{e(X_i)}\right] - \mathbb{E}\left[\tau(X_i)\frac{(1-e(X_i))\mathbb{E}\left[Y_i(0) - \mu_0(X_i) \mid X_i\right]}{1-e(X_i)}\right] \\
&= 0
\end{aligned}
$$

Then,

$$
\mathbb{V}\left[\hat{\tau}^{\mathrm{AIPW}^*}\right] = \mathbb{V}\left[\frac{1}{n}\sum_{i=1}^{n}\left(\mu_1(X_i) - \mu_0(X_i) + \frac{W_i(Y_i - \mu_1(X_i))}{e(X_i)} - \frac{(1 - W_i)(Y_i - \mu_0(X_i))}{1 - e(X_i)}\right)\right]
$$

$$
= \frac{1}{n^2}\sum_{i=1}^{n}\left(\mathbb{V}\left[\tau(X_i)\right] + \mathbb{V}\left[\frac{W_i(Y_i - \mu_1(X_i))}{e(X_i)}\right] + \mathbb{V}\left[\frac{(1 - W_i)(Y_i - \mu_0(X_i))}{1 - e(X_i)}\right] + 2A\right)
$$

$$
= \frac{1}{n^2}\sum_{i=1}^{n}\left(\mathbb{V}\left[\tau(X_i)\right] + \mathbb{E}\left[\left(\frac{(Y - \mu_1(X_i))^2}{e(X_i)}\right)\right] + \mathbb{E}\left[\left(\frac{(Y - \mu_0(X_i))^2}{1 - e(X_i)}\right)^2\right]\right)
$$

$$
= \frac{1}{n}\left(\mathbb{V}\left[\tau(X_i)\right] + \mathbb{E}\left[\left(\frac{(Y - \mu_1(X_i))^2}{e(X_i)}\right)\right] + \mathbb{E}\left[\left(\frac{(Y - \mu_0(X_i))^2}{1 - e(X_i)}\right)^2\right]\right)
$$

Because these variances are well defined from what precedes, the Central Limit Theorem can be applied to $\hat{\tau}^{\mathrm{IPW}^*}$ and $\hat{\tau}^{\mathrm{AIPW}^*}$, which gives the result in Theorem 1. $\square$

### A.3 Proof of Theorem 2

*Proof.* **Unbiasedness.** We first prove the unbiasedness of the local IPW estimators:

$$
\mathbb{E}\left[\hat{\tau}_k^{\mathrm{IPW}^*} \mid H_i = k\right] = \mathbb{E}\left[\frac{W_iY_i}{e_k(X_i)} - \frac{(1 - W_i)Y_i}{1 - e_k(X_i)} \mid H_i = k\right] \qquad \text{i.i.d.}
$$

$$
= \mathbb{E}\left[\frac{W_iY_i(1)}{e_k(X_i)} - \frac{(1 - W_i)Y_i(0)}{1 - e_k(X_i)} \mid H_i = k\right] \qquad \text{SUTVA}
$$

$$
= \mathbb{E}\left[\mathbb{E}\left[\frac{W_iY_i(1)}{e_k(X_i)} \mid H_i = k, X_i\right] - \mathbb{E}\left[\frac{(1 - W_i)Y_i(0)}{1 - e_k(X_i)} \mid X_i, H_i = k, X_i\right]\right]
$$

$$
= \mathbb{E}\left[\frac{\mathbb{E}\left[W_i \mid H_i = k, X_i\right]\mathbb{E}\left[Y_i(1) \mid H_i = k, X_i\right]}{e_k(X_i)}\right.
$$

$$
\left. - \frac{\mathbb{E}\left[(1 - W_i) \mid H_i = k, X_i\right]\mathbb{E}\left[Y_i(0) \mid H_i = k, X_i\right]}{1 - e_k(X_i)}\right] \qquad \text{Ass. 1}
$$

$$
= \mathbb{E}\left[\mathbb{E}\left[Y_i(1) - Y_i(0) \mid H_i = k, X_i\right]\right]
$$

$$
= \mathbb{E}\left[Y_i(1) - Y_i(0) \mid H_i = k\right]
$$

$$
= \tau_k
$$

This yields

$$
\mathbb{E}\left[\hat{\tau}^{\mathrm{meta\text{-}IPW}^*}\right] = \mathbb{E}\left[\sum_{k=1}^{K}\frac{n_k}{n}\hat{\tau}_k^{\mathrm{IPW}^*}\right]
$$

$$
= \sum_{k=1}^{K}\mathbb{E}\left[\frac{n_k}{n}\mathbb{E}\left[\hat{\tau}_k^{\mathrm{IPW}^*} \mid H_i = k\right]\right]
$$

$$
= \sum_{k=1}^{K}\mathbb{E}\left[\frac{n_k}{n}\right]\tau_k
$$

$$
= \sum_{k=1}^{K}\rho_k\tau_k
$$

$$
= \tau
$$

**Variance.** For the variance of the local ATEs, we follow the same steps as in the previous proof which yields

$$\mathbb{V}\left[\hat{\tau}_k^{\text{IPW}^*} \mid H = k\right] = \frac{1}{n_k}\left(\mathbb{E}\left[\frac{Y_i(1)^2}{e_k(X_i)} \mid H_i = k\right] - \mathbb{E}\left[\frac{Y_i(0)^2}{1 - e_k(X_i)} \mid H_i = k\right] - \tau_k^2\right)$$

$$= \frac{1}{n_k}V_{k,i}$$

with $V_{k,i} = \mathbb{E}\left[\frac{Y_i(1)^2}{e_k(X_i)} \mid H_i = k\right] - \mathbb{E}\left[\frac{Y_i(0)^2}{1 - e_k(X_i)} \mid H_i = k\right] - \tau_k^2$. Finally, by Lemma 2, we have

$$\mathbb{V}\left[\hat{\tau}^{\text{meta-IPW}^*}\right] = \mathbb{E}\left[\mathbb{V}\left[\hat{\tau}^{\text{meta-IPW}^*} \mid H = k\right]\right] + \mathbb{V}\left[\mathbb{E}\left[\hat{\tau}^{\text{meta-IPW}^*} \mid H = k\right]\right]$$

$$= \frac{1}{n}\sum_{k=1}^{K} \rho_k V_{k,i} + \frac{1}{n}\mathbb{V}\left[\tau_H\right]$$

The proof for the meta AIPW estimator follows the same steps. □

## A.4 DECOMPOSITION OF THE GLOBAL PROPENSITY SCORE

By the law of total probabilities,

$$e(x) = \mathbb{P}(W_i = 1 \mid X_i)$$

$$= \sum_{k=1}^{K} \mathbb{P}(W_i = 1 \cap H_i = k \mid X_i = x)$$

$$= \sum_{k=1}^{K} \mathbb{P}(H_i = k \mid X_i)\mathbb{P}(W_i = 1 \mid X_i = x, H_i = k)$$

$$= \sum_{k=1}^{K} \mathbb{P}(H_i = k)\frac{P(X_i \mid H_i = k)}{P(X_i)}e_k(X_i) \qquad \text{(DW)}$$

$$= \sum_{k=1}^{K} \mathbb{P}(H_i = k \mid X_i)e_k(X_i) \qquad \text{(MW)}$$

## A.5 PROOF OF THEOREM 3

*Proof.*

$$\hat{\tau}_{\text{IPW}}^{\text{fed}^*} = \sum_{k=1}^{K} \frac{n_k}{n}\hat{\tau}_{\text{IPW}}^{\text{fed}(k)}$$

$$= \sum_{k=1}^{K} \frac{n_k}{n}\left(\frac{1}{n_k}\sum_{i=1}^{n_k}\left(\frac{W_i Y_i}{e(X_i)} - \frac{(1 - W_i)Y_i}{1 - e(X_i)}\right)\right)$$

$$= \frac{1}{n}\sum_{i=1}^{n}\left(\frac{W_i Y_i}{e(X_i)} - \frac{(1 - W_i)Y_i}{1 - e(X_i)}\right)$$

$$= \hat{\tau}^{\text{IPW}^*}$$

We prove similarly that $\hat{\tau}_{\text{AIPW}}^{\text{fed}^*} = \hat{\tau}^{\text{AIPW}^*}$. □

## A.6 PROOF OF THEOREM 4

We start by two technical lemmas.

**Lemma 1.** *Under Assumptions 1 and 2 we have,*

$$\mathbb{V}\left[\tau(X_i)\right] = \sum_{k=1}^{K} \rho_k \mathbb{V}\left[\tau(X_i) \mid H_i = k\right] + \mathbb{V}\left[\tau_{H_i}\right]$$

*Proof.*

$$\mathbb{V}\left[\tau(X_i)\right] = \mathbb{E}\left[\mathbb{V}\left[\tau(X_i) \mid H_i = k\right]\right] + \mathbb{V}\left[\mathbb{E}\left[\tau(X_i) \mid H_i = k\right]\right]$$

$$= \sum_{k=1}^{K} \rho_k \mathbb{V}\left[\tau(X_i) \mid H_i = k\right] + \mathbb{V}\left[\sum_{k=1}^{K} \mathbb{1}_{[H_i=k]} \tau_{H_i}\right]$$

$$= \sum_{k=1}^{K} \rho_k \mathbb{V}\left[\tau(X_i) \mid H_i = k\right] + \mathbb{V}\left[\tau_{H_i} \sum_{k=1}^{K} \mathbb{1}_{[H_i=k]}\right]$$

$$= \sum_{k=1}^{K} \rho_k \mathbb{V}\left[\tau(X_i) \mid H_i = k\right] + \mathbb{V}\left[\tau_{H_i}\right]$$

$\square$

**Lemma 2.** *For the general form of meta-analysis estimator $\hat{\tau}^{\mathrm{meta}} = \sum_{k=1}^{K} \frac{n_k}{n} \hat{\tau}_k$, we have*

$$\mathbb{V}\left[\hat{\tau}^{\mathrm{meta}}\right] = \frac{1}{n} \sum_{k=1}^{K} \rho_k V_k + \frac{1}{n} \mathbb{V}\left[\tau_{H_i}\right]$$

*where $V_k = \mathbb{V}\left[\hat{\tau}_k \mid H_i = k\right]$ is the within-site variance of the ATE estimator in site $k$ and $\mathbb{V}\left[\tau_{H_i}\right] = \mathbb{V}\left[\mathbb{E}\left[Y_i(1) - Y_i(0) \mid H_i\right]\right]$ is the between-sites variance of the local ATEs.*

*Proof.*

$$\mathbb{V}\left[\hat{\tau}^{\mathrm{meta}}\right] = \mathbb{V}\left[\sum_{k=1}^{K} \frac{n_k}{n} \hat{\tau}_k\right]$$

$$= \mathbb{E}\left[\mathbb{V}\left[\sum_{k=1}^{K} \frac{n_k}{n} \hat{\tau}_k \mid H_1, \ldots, H_n\right]\right] + \mathbb{V}\left[\mathbb{E}\left[\sum_{k=1}^{K} \frac{n_k}{n} \hat{\tau}_k \mid H_1, \ldots, H_n\right]\right]$$

$$= \mathbb{E}\left[\sum_{k=1}^{K} \frac{n_k}{n^2} V_k\right] + \mathbb{V}\left[\sum_{k=1}^{K} \frac{n_k}{n} \tau_k\right] = \frac{1}{n} \sum_{k=1}^{K} \rho_k V_k + \mathbb{V}\left[\frac{1}{n} \sum_{k=1}^{K} \sum_{i=1}^{n} \mathbb{1}_{[H_i=k]} \tau_{H_i}\right]$$

$$= \frac{1}{n} \sum_{k=1}^{K} \rho_k V_k + \mathbb{V}\left[\frac{1}{n} \sum_{i=1}^{n} \tau_{H_i} \sum_{k=1}^{K} \mathbb{1}_{[H_i=k]}\right] = \frac{1}{n} \sum_{k=1}^{K} \rho_k V_k + \mathbb{V}\left[\frac{1}{n} \sum_{i=1}^{n} \tau_{H_i}\right]$$

$$= \frac{1}{n} \sum_{k=1}^{K} \rho_k V_k + \frac{1}{n} \mathbb{V}\left[\tau_H\right],$$

$\square$

We can now prove Theorem 4.

*Proof.* We begin with IPW estimators, and then move to AIPW.

**IPW.** First, applying Jensen's inequality with the strictly convex function $t \mapsto \frac{1}{t}$ in $]0; 1[$ and summing-to-one weights $\omega_k(X) = \mathbb{P}(H_i = k \mid X_i = X)$, we have

$$\mathbb{E}\left[\frac{Y_i(1)^2}{e(X_i)}\right] < \mathbb{E}\left[\sum_{k=1}^{K} \omega_k(X) \frac{Y_i(1)^2}{e_k(X_i)}\right]$$

if $\exists (k, k') \in [K], e_k(X_i) \neq e_{k'}(X_i)$, and equality if $\forall k \in [K], e_k(X_i) = e(X_i)$, i.e. if the local propensity scores are all equal to one another. Then, with the same condition on strictness and equality,

$$\mathbb{E}\left[\frac{Y_i(1)^2}{e(X_i)}\right] \leq \mathbb{E}\left[\sum_{k=1}^{K} \omega_k(X) \frac{Y_i(1)^2}{e_k(X_i)}\right]$$

$$= \sum_{k=1}^{K} \mathbb{E}\left[\mathbb{E}\left[\mathbb{1}_{[H_i=k]} \mid X_i\right] \frac{Y_i(1)^2}{e_k(X_i)}\right]$$

$$= \sum_{k=1}^{K} \mathbb{E}\left[\mathbb{E}\left[\mathbb{1}_{[H_i=k]} \frac{Y_i(1)^2}{e_k(X_i)} \mid X_i\right]\right]$$

$$= \sum_{k=1}^{K} \mathbb{E}\left[\mathbb{1}_{[H_i=k]} \frac{Y_i(1)^2}{e_k(X_i)}\right]$$

$$= \sum_{k=1}^{K} \rho_k \mathbb{E}\left[\frac{Y_i(1)^2}{e_k(X_i)} \mid H_i = k\right]$$

Similarly, $\mathbb{E}\left[\frac{Y_i(0)^2}{1-e(X_i)}\right] \leq \sum_{k=1}^{K} \rho_k \mathbb{E}\left[\frac{Y_i(0)^2}{1-e_k(X_i)} \mid H_i = k\right]$. Then,

$$\mathbb{V}\left[\hat{\tau}^{\text{fed-IPW}^*}\right] = \frac{1}{n}\left(\mathbb{E}\left[\frac{Y_i(1)^2}{e(X_i)} + \frac{Y_i(0)^2}{1-e(X_i)}\right] - \tau^2\right)$$

$$\leq \frac{1}{n}\left(\sum_{k=1}^{K} \rho_k \mathbb{E}\left[\frac{Y_i(1)^2}{e_k(X_i)} + \frac{Y_i(0)^2}{1-e_k(X_i)} \mid H_i = k\right] - \tau^2\right)$$

$$\leq \frac{1}{n}\left(\sum_{k=1}^{K} \rho_k \underbrace{\left(\mathbb{E}\left[\frac{Y_i(1)^2}{e_k(X_i)} + \frac{Y_i(0)^2}{1-e_k(X_i)} \mid H_i = k\right] - \tau_k^2\right)}_{:=V_k} + \underbrace{\sum_{k=1}^{K} \rho_k \tau_k^2 - \tau^2}_{\mathbb{V}(\tau_H)}\right)$$

On the other hand, by Lemma 2,

$$\mathbb{V}\left[\hat{\tau}^{\text{meta-IPW}^*}\right] = \frac{1}{n}\sum_{k=1}^{K} \rho_k V_k + \frac{1}{n}\mathbb{V}\left[\tau_H\right].$$

Hence $\mathbb{V}\left[\hat{\tau}^{\text{fed-IPW}^*}\right] = \mathbb{V}\left[\hat{\tau}^{\text{meta-IPW}^*}\right]$ if $\forall k \in [K], e_k = e$, and $\mathbb{V}\left[\hat{\tau}^{\text{fed-IPW}^*}\right] < \mathbb{V}\left[\hat{\tau}^{\text{meta-IPW}^*}\right]$ if $\exists (k, k') \in [K]^2, e_k \neq e_{k'}$.

**AIPW.** Similarly to the IPW case, we have

$$\mathbb{E}\left[\left(\frac{W_i(Y_i - \mu_1)}{e(X_i)}\right)^2\right] \leq \sum_{k=1}^{K} \rho_k \mathbb{E}\left[\left(\frac{W_i(Y_i - \mu_1)}{e_k(X_i)}\right)^2 \mid H_i = k\right],$$

$$\mathbb{E}\left[\left(\frac{(1-W_i)(Y_i - \mu_0)}{1-e(X_i)}\right)^2\right] \leq \sum_{k=1}^{K} \rho_k \mathbb{E}\left[\left(\frac{(1-W_i)(Y_i - \mu_0)}{1-e_k(X_i)}\right)^2 \mid H_i = k\right],$$

and with Lemma 1:

$$\mathbb{V}\left[\tau(X_i)\right] = \sum_{k=1}^{K} \rho_k \mathbb{V}\left[\tau(X_i) \mid H_i = k\right] + \mathbb{V}\left[\tau_{H_i}\right].$$

Then, using Lemma 2, we have the desired result. $\qquad \square$

### A.7 PROOF OF THEOREM 5

*Proof.* Let $f : t \mapsto \frac{1}{t(1-t)}$ with $t \in ]0,1[$. $f$ is convex. Then by Jensen's inequality with summing-to-one weights $\omega_k(X) = \mathbb{P}(H_i = k \mid X_i = X)$,

$$
\mathcal{O}_{\text{global}} = \mathbb{E}\left[ f\left( \sum_{k=1}^{K} \omega_k(X) e_k(X) \right) \right]
$$

$$
\leq \mathbb{E}\left[ \sum_{k=1}^{K} \omega_k(X) f(e_k(X)) \right]
$$

$$
\leq \sum_{k=1}^{K} \mathbb{E}\left[ \mathbb{E}\left[ \mathbb{1}_{[H_i=k|X_i]} \right] f(e_k(X)) \right]
$$

$$
\leq \sum_{k=1}^{K} \mathbb{E}\left[ \mathbb{E}\left[ \mathbb{1}_{[H_i=k]} f(e_k(X)) \mid X_i \right] \right]
$$

$$
\leq \sum_{k=1}^{K} \mathbb{E}\left[ \mathbb{1}_{[H_i=k]} f(e_k(X)) \right]
$$

$$
\leq \sum_{k=1}^{K} \mathbb{P}(H_i = k) \mathbb{E}\left[ f(e_k(X)) \mid H_i = k \right]
$$

$$
\leq \sum_{k=1}^{K} \rho_k \mathcal{O}_k \qquad \square
$$

## B FEDERATED LEARNING OF MEMBERSHIP WEIGHTS

We describe how to estimate membership weights using a general parametric classification model trained via Federated Averaging (FedAvg) (McMahan et al., 2017), without requiring access to individual-level data.

### B.1 GENERAL PARAMETRIC MODEL

Let $\omega_k(X_i; \Theta)$ denote the predicted membership probability of site $k$ for covariate vector $X_i$, where $\Theta$ are the parameters of the model. For a generic parametric model, $\Theta \in \mathbb{R}^p$ represents all trainable parameters, and the membership-weight estimation problem can be formulated as the minimization of the empirical cross-entropy loss:

$$
\ell(\Theta; \mathcal{D}) = -\frac{1}{|\mathcal{D}|} \sum_{i=1}^{|\mathcal{D}|} \sum_{k=1}^{K} \mathbb{1}_{\{H_i=k\}} \log\big(\omega_k(X_i; \Theta)\big),
$$

where $H_i^{\text{enc}}$ is the one-hot encoding of the site membership $H_i$.

This framework encompasses a broad class of models, including:

- Multinomial logistic regression:

$$
\omega_k(X_i; \Theta) = \frac{\exp(\theta_k^\top X_i)}{\sum_{k'=1}^{K} \exp(\theta_{k'}^\top X_i)}, \quad \Theta = (\theta_1, \ldots, \theta_K) \in \mathbb{R}^{d \times K}.
$$

- Neural networks, where $\Theta$ includes all weights and biases across layers, and $\omega_k(\cdot)$ is the softmax output of the final layer.

### B.2 FEDERATED TRAINING PROCEDURE

Algorithm 1 summarizes the generic Federated Averaging procedure for training any differentiable parametric model for membership-weight estimation.

---

**Algorithm 1** Federated Learning of Membership Weights with a Parametric Model

---

1: **Input:** $K$ sites, $E$ local steps, $\eta$ learning rate, $T$ communication rounds, $B$ batch size
2: Initialize global parameters $\Theta_0$
3: **for** $t = 1$ to $T$ **do**
4:     **for** each client $k \in [1, \ldots, K]$ **in parallel do**
5:         $\Theta_{t+1}^{(k)} \leftarrow \text{LOCALUPDATE}(k, \Theta_t)$
6:     **end for**
7:     $\Theta_{t+1} \leftarrow \sum_{k=1}^{K} \frac{n_k}{n} \Theta_{t+1}^{(k)}$                  // FedAvg aggregation
8: **end for**
9: **LocalUpdate**$(k, \Theta)$:
10: **for** $e = 1$ to $E$ **do**
11:     Sample mini-batch $\mathcal{B}_k$ of size $B$ from local data $\mathcal{D}_k$
12:     Compute predictions $\omega_k(X_i; \Theta)$ for $i \in \mathcal{B}_k$
13:     Compute gradient $\nabla \ell(\Theta; \mathcal{B}_k)$ of cross-entropy loss
14:     $\Theta \leftarrow \Theta - \eta \nabla \ell(\Theta; \mathcal{B}_k)$
15: **end for**
16: **return** $\Theta$

---

With a suitable choice of learning rate $\eta$ and a moderate number of local steps $E$ per round, Algorithm 1 converges to the same solution as centralized training as the number of communication rounds $T \to \infty$ (Stich, 2019; Khaled et al., 2020; Li et al., 2019). The resulting federated parameter estimate $\widehat{\Theta}^{\text{fed}}$ yields the estimated membership weights:

$$\hat{\omega}_k(X_i) = \omega_k(X_i; \widehat{\Theta}^{\text{fed}}), \qquad k = 1, \ldots, K.$$

### B.3 EXAMPLE: MULTINOMIAL LOGISTIC REGRESSION

When the model is multinomial logistic regression, $\Theta$ is the matrix of regression coefficients, and the local gradient takes the closed form:

$$\nabla \ell(\Theta; \mathcal{B}_k) = \frac{1}{B} \sum_{i \in \mathcal{B}_k} X_i \big( \omega(X_i; \Theta) - H_i^{\text{enc}} \big),$$

where $\omega(X_i; \Theta)$ is the vector of softmax probabilities for sample $i$.

## C SIMULATION DETAILS

The parameters common to all settings are shown in Table 1, where $\gamma_2^{(\text{weak})} = [-2.5, -1, -0.15, -0.15, 0, -0.15, -1, -0.15, -0.15, 0]$ and $\gamma_2^{(\text{good})} = [-.05, -.1, -.05, -.1, .05, -.1, -.05, -.1, .05, -.1]$.

| Parameter | Center 1 | Center 2 | Center 3 |
|:---:|:---:|:---:|:---:|
| $d$ | 10 | | |
| $\mu_1(X)$ | $\sum_{j=1}^{5} \frac{j}{10} X_j^2 + \sum_{j=6}^{10} \frac{j}{10} X_j + X_9 * X_{10}$ | | |
| $\mu_0(X)$ | $\sum_{j=1}^{5} \frac{3j-10}{30} X_j^2 + \sum_{j=6}^{10} \frac{3j-10}{30} X_j + X_1 * X_{10}$ | | |
| $e_k$ | Logistic$(x, \gamma_k)$ | | |
| $\gamma_k$ | $[-.25, .25, -.25, -.25, .25, -.25, -.25, .25, -.25, .25]$ | No overlap: not logistic (only controls) 

 Weak overlap: $\gamma_2^{(\text{weak})}$ 
 Good overlap: $\gamma_2^{(\text{good})}$ | $[.15, -.15, .15, -.15, .15, -.15, .15, -.15, .15, -.15]$ |

Table 1: Common simulation parameters.

DGP A-specific settings are shown in Table 2, where $J_d$ is the $d \times d$ matrix of ones, and $I_d$ is the $d \times d$ identity matrix.

| Parameter | Center 1 | Center 2 | Center 3 |
|---|---|---|---|
| $n_k$ | 650 | | |
| $\mathcal{D}_k$ | $\mathcal{N}(\mu_k, \Sigma_k)$ | | |
| $\mu_k$ | $(1, \ldots, 1) \in \mathbb{R}^d$ | $(1.5, 1.5, 1.5, 1, \ldots, 1) \in \mathbb{R}^d$ | $(2, 2, 2, 1, \ldots, 1) \in \mathbb{R}^d$ |
| $\Sigma_k$ | $I_d + 0.5 J_d$ | $0.6 I_d + 0.4 J_d$ | $3 I_d + 0.3 J_d$ |

Table 2: Simulation parameters specific to DGP A.

DGP B-specific settings are shown in Table 3.

| Parameter | Value |
|---|---|
| $n$ | 4000 |
| $\mathcal{D}$ | $\frac{2}{3}\mathcal{N}(\mu_1, \Sigma_1) + \frac{1}{3}\mathcal{N}(\mu_2, \Sigma_2)$ |
| $\mu_1$ | $(0, \ldots, 0) \in \mathbb{R}^d$ |
| $\mu_2$ | $(1.5, \ldots, 1.5) \in \mathbb{R}^d$ |
| $\Sigma_1$ | $I_d$ |
| $\Sigma_2$ | $I_d + 0.5 J_d$ |
| $\mathbb{P}(H_i = k \mid X)$ | $\mathrm{Logistic}(x, \theta_k)$ |
| $\theta_1$ | $[-0.5, -0.5, 0.2, -0.5, -0.5, 0.2, -0.5, -0.5, 0.2, 0.2]$ |
| $\theta_2$ | $[0.5, 0.5, 0.2, 0.5, 0.5, 0.2, 0.5, 0.5, 0.2, 0.5]$ |
| $\theta_3$ | $[1, 1, 0.2, 0.2, 0.2, 0.2, 0.2, 0.2, 0.2, 0.2]$ |

Table 3: Simulation parameters specific to DGP B.

**Real data.** We classified a site as 'large' if it had at least $5 \times d$ observations in each treatment arm (minimum 85 per arm in our setting). For such sites, we estimated $e_k$ with probability forests with $2,000$ trees; otherwise, we used logistic regression. We estimated Fed-MW RF with a centralized random forest for $P(H|X)$ with 500 trees.

# D    OVERLAP IMPROVEMENT

Echoing theorem 5, Figures 5 and 6 display the empirical distributions (on a log-scale) of the local propensity score in site 2 ($e_2$) and of the global propensity score $e$ in the *Poor local overlap* scenario (see Figure 2b for corresponding results). A good overlap is when both the propensity score distributions for the treated and control overlap, and are far from 0. We see that the poor overlap at site 2 (Figure 5), with values of $e_2$ on the local data close to 0, is significantly improved at the global level (Figure 6).

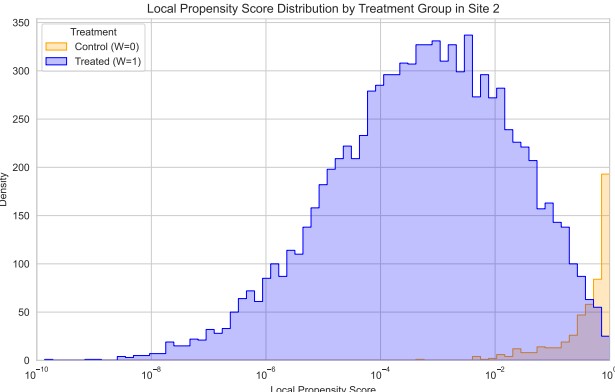

Figure 5: Local overlap in site 2 for the *Poor local overlap* scenario (DGP A).

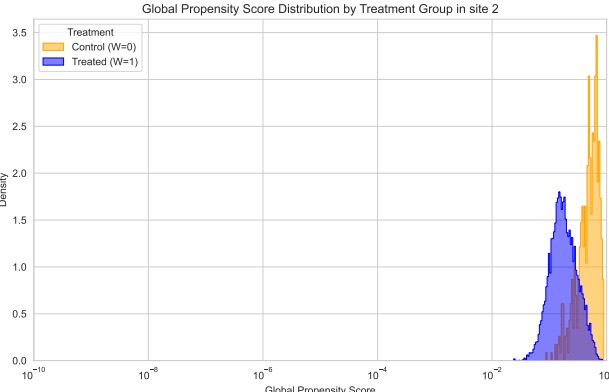

Figure 6: Global overlap for the *Poor local overlap* scenario (DGP A). We see a clear improvement compared to the local overlap in site 2 (Figure 5).

# E    ADDITIONAL SIMULATIONS

## E.1    LARGE NUMBER OF SITES

We further investigate the robustness of our estimators as the number of sites increases. Specifically, we simulate $K = 20$ sites under weak local overlap (corresponding to the setting of Figure 2b in the main paper) and report the bias and mean squared error (MSE) for both data-generating processes (DGP A and DGP B).

Table 4: Bias and mean squared error (MSE) under $K = 20$ sites with weak local overlap. Bold values indicate unbiasedness or lowest MSE.

| Method | Bias (DGP A) | MSE (DGP A) | Bias (DGP B) | MSE (DGP B) |
|---|---|---|---|---|
| Oracle IPW | 0.00 | 0.298 | 0.00 | 0.130 |
| Fed. IPW-MW (logistic) | 0.48 | 0.294 | **0.00** | **0.095** |
| Fed. IPW-MW (NN) | **0.00** | 0.240 | **0.00** | 0.123 |
| Fed. IPW-DW | **0.00** | **0.096** | 0.50 | 0.450 |
| Meta-SW IPW | -2.09 | 13.279 | -1.77 | 7.977 |
| Oracle AIPW | 0.00 | 0.007 | 0.00 | 0.049 |
| Fed. AIPW-MW (logistic) | 0.79 | 0.639 | **0.00** | **0.078** |
| Fed. AIPW-MW (NN) | **0.00** | 0.153 | **0.00** | 0.084 |
| Fed. AIPW-DW | **0.00** | **0.019** | 0.25 | 0.194 |
| Meta-SW AIPW | -1.24 | 1.775 | -2.36 | 18.465 |

As shown in Table 4, estimators using neural network membership weights consistently achieve unbiasedness across both DGPs, although with slightly higher variance compared to their well-specified parametric counterparts. This is consistent with the behavior of more flexible models, which trade off a small increase in variance for robustness to misspecification. When parametric assumptions are correctly specified (e.g., Gaussian density weights in DGP A, logistic membership weights in DGP B), these methods achieve the lowest MSE. Overall, the NN-based approach demonstrates robust consistency across DGPs and scales well with the number of sites.

### E.2 FULLY MODEL-AGNOSTIC ATE ESTIMATION

We additionally investigate the benefits of fully model-agnostic estimation for membership weights $P(H \mid X)$. In this setting, we use federated neural networks to estimate membership probabilities in a flexible way and pair them with non-parametric local propensity score estimators (e.g., Random Forests). This procedure does not require prior parametric knowledge about $H \mid X$ or $X \mid H$, making it robust to a wide range of DGPs.

We simulate $K = 3$ sites with $d = 10$ covariates, $n_k = 600$ observations per site, and $X \mid H = k \sim D_k$ as a mixture of Gaussians with weak local overlap. The results are summarized in Table 5.

| Method | Bias | MSE |
|---|---|---|
| Oracle IPW | 0.00 | 0.116 |
| **Fed. IPW-MW (NN MW + RF $e_k$)** | **0.00** | **0.075** |
| Meta-SW IPW (RF $e_k$) | 0.89 | 0.884 |
| Oracle AIPW | 0.00 | 0.030 |
| **Fed. AIPW-MW (NN MW + RF $e_k$)** | **0.00** | **0.040** |
| Meta-SW AIPW (RF $e_k$) | -0.36 | 0.180 |

Table 5: Bias and MSE under fully non-parametric estimation of membership weights and local propensity scores ($K = 3$, weak local overlap). Bold values indicate unbiased (or nearly unbiased) estimators.

As shown in Table 5, our federated approach with neural-network-based membership weights and random-forest local propensities yields nearly unbiased estimates with substantially lower MSE compared to meta-analysis estimators. This demonstrates that misspecifications in either the membership weights or the local propensity scores can be effectively mitigated by adopting fully non-parametric methods in a federated learning setting.

### E.3 Robustness to local misspecifications

Similar to meta-analysis estimators, we argue that inconsistencies in local propensity models can be mitigated as $K$ grows, provided that such inconsistencies are marginal relative to the number of correctly specified sites. To illustrate this, we simulate $K$ sites under DGP B, each holding $n_k = 60$ observations drawn from homogeneous covariate distributions and sharing the same true propensity model (logistic). We then intentionally misspecify the propensity model for site $k = 1$, setting $\widehat{e}_1(x) \equiv 0.1$ for all $x$.

Table 6 reports the bias and mean squared error (MSE) of several estimators under two scenarios: $K = 3$ (few sites) and $K = 200$ (many sites).

| **Method** | **Bias** ($K = 3$) | **MSE** ($K = 3$) | **Bias** ($K = 200$) | **MSE** ($K = 200$) |
|---|---|---|---|---|
| Oracle IPW | 0.00 | 0.272 | 0.00 | 0.002 |
| Fed. IPW (logistic MW) | 0.38 | 0.344 | 0.00 | 0.002 |
| Meta IPW | 2.61 | 7.902 | 0.03 | 0.003 |
| Oracle AIPW | 0.00 | 0.058 | 0.00 | 0.001 |
| Fed. AIPW (logistic MW) | -0.33 | 0.325 | -0.01 | 0.002 |
| Meta AIPW | -0.51 | 0.481 | -0.07 | 0.007 |

Table 6: Bias and mean squared error (MSE) under local misspecification of site $k = 1$ propensity model.

With only $K = 3$ sites, both federated and meta-analysis estimators exhibit substantial bias. However, as the number of sites increases, the influence of the single misspecified site becomes negligible, and the bias decreases markedly (e.g., from 0.38 to 0.00 for Federated IPW with logistic membership weights). As expected, the AIPW estimator retains its double-robustness property even under misspecification of the outcome model, leading to consistently low bias and MSE across settings.

## F    Extension to Site Effects

In this section, we show how our approach can be adapted to handle site-effects by testing and relaxing Assumption 2.

### F.1    Federated tests for site effects

Assumption 2 posits that site membership $H$ provides no information about potential outcomes conditional on covariates $X$ and treatment $W$. This assumption is *testable* by fitting a federated regression model of $Y$ against $(X, H)$, and testing whether the coefficients for $H$ are jointly zero. In federated learning, this test is implemented via one-shot sharing of sufficient statistics: each site transmits its local covariance matrix $\hat{\Sigma}_k = \frac{1}{n_k} \sum_{i=1}^{n_k} X_i^\top X_i$ and vector $\hat{\gamma}_k = \frac{1}{n_k} \sum_{i=1}^{n_k} X_i^\top Y_i$. The server aggregates these to compute a joint F-test or Wald test on the $H$ block, yielding a $p$-value. We recommend adapting the estimation strategy based on this test's result (see Figure **??**).

### F.2    Handling site effects through site-level covariates

Site effects are often driven by observable site-level covariates $Z$ (e.g., equipment quality, staff availability). In such scenarios, the causal structure is augmented by the path $H \rightarrow Z \rightarrow Y$, as shown in Figure 7. Under this decomposition, a weaker, testable alternative to Assumption 2 suffices for identification.

**Assumption 5** (Explainable site effects)**.** *The potential outcomes are independent of site membership given individual and site-level covariates:*

$$\{Y(1), Y(0)\} \perp H \mid (X, Z).$$

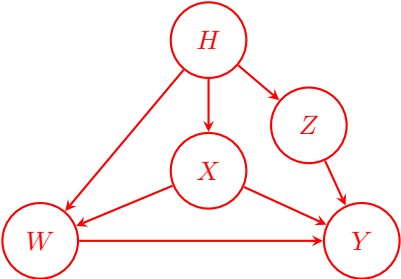

Figure 7: Graphical model for multi-site observational data with site covariates $Z$ mediating the site effect.

**Adapting our estimators.** Under Assumption 5, one needs to modify the global propensity score to account for $Z$:

$$e(X, Z) \;=\; \sum_{k=1}^{K} P(H=k \mid X, Z) \, e_k(X).$$

Our MW framework accommodates this extension seamlessly: it identifies the effect by learning membership weights $P(H \mid X, Z)$ using both $X$ and $Z$, and by similarly updating the outcome models of Federated AIPW to $\mu_w(X, Z)$. Importantly, local propensity scores $e_k(X)$ remain functions of $X$, simplifying local computation (since $Z$ is constant within sites). Note however that we now need to extend Assumption 3 into

$$\mathcal{O}_{\text{global}} = \mathbb{E}\left[ (e(X,Z)(1-e(X,Z)))^{-1} \right] < +\infty.$$

**Experiments.** To validate this approach, we simulate $K = 30$ sites, each with $n_k \in \{30, 90, 150\}$ observations. Covariates $X \in \mathbb{R}^3 \sim \mathcal{N}(0, I_3)$ are drawn i.i.d. and allocated to sites via $H \sim \text{Logit}(\theta^\top X)$ to induce covariate shift. Site-level covariates $Z \in \mathbb{R}^3$ (site effects) are drawn from site-specific Gaussians: for each site $k$, a centroid $\mu_k \sim \mathcal{N}([1,1,1], I_3)$ is drawn, then $Z \mid H = k \sim \mathcal{N}(\mu_k, I_3)$. Treatment is assigned by site-specific logistic models. Outcomes are linear: $Y = X^\top \beta + Z^\top \delta + \tau W + \varepsilon$, with $\varepsilon \sim \mathcal{N}(0,1)$, $\tau = 10$, and $\beta = \delta = [-2, -1.33, -0.66]$.

We estimate membership probabilities $P(H \mid \cdot)$ using multinomial logistic regression or a two-layer neural network (NN), trained on either $(X)$ or $(X, Z)$. Local propensities $\hat{e}_k(X)$ are fitted via logistic regression, and federated outcome models $\hat{\mu}_w(X, Z)$ via OLS. Membership weights are aggregated to form $\hat{e}(x, z) = \sum_k \hat{P}(H=k \mid x, z)\hat{e}_k(x)$. IPW and AIPW estimators are computed as $\hat{\tau}_{\text{IPW}} = \frac{1}{n} \sum_i \frac{W_i Y_i}{\hat{e}(X_i)} - \frac{1}{n} \sum_i \frac{(1-W_i)Y_i}{1-\hat{e}(X_i)}$ and $\hat{\tau}_{\text{AIPW}} = \frac{1}{n} \sum_i \left[ \frac{W_i(Y_i - \hat{\mu}_1)}{\hat{e}(X_i)} - \frac{(1-W_i)(Y_i - \hat{\mu}_0)}{1-\hat{e}(X_i)} + (\hat{\mu}_1 - \hat{\mu}_0) \right]$. Meta-analysis estimators adjust only for $X$, as averaging over $H$ inherently stratifies by site.

The results shown in Table 7 confirm that adjusting via $\hat{e}(x, z)$ effectively uses $Z$ as a proxy for the latent site indicator $H$. By basing membership weights on $(X, Z)$, we remove confounding from both covariate shift ($H \to X$) and site effects ($H \to Z \to Y$) without incorrectly including $Z$ in local propensity models. Consequently, estimators adjusted for $(X, Z)$ are unbiased, whereas those adjusted only for $X$ remain biased. Federated AIPW with $(X, Z)$ achieves very low MSE, as the well-specified outcome model $\hat{\mu}_w(X, Z)$ corrects for residual imbalances. Meta-analysis estimators perform poorly with small $n_k$ due to high variance in local nuisance estimation, whereas federated estimators borrow strength across sites to achieve lower MSE.

### F.3 UNEXPLAINED SITE EFFECTS

If site-level covariates $Z$ are unavailable or insufficient to explain all site effects (e.g., due to a direct $H \to Y$ edge, see Figure 8), the core identification assumption remains Assumption 1: $\{Y(1), Y(0)\} \perp W \mid (X, H))$. In this regime, nuisance functions—propensity scores $e(X, Z, H)$ and outcome models $\mu_w(X, Z, H)$—must be learned federatively using parametric models that explicitly include site fixed effects ($H$). Including $Z$ in these models helps to reduce the variance of the nuisance functions. Note that the theoretical and practical advantage of federated learning over meta-analysis is diminished in this setting, as meta estimation effectively stratifies inference on $H$.

Table 7: Mean Squared Error (MSE) of treatment effect estimators across varying site sizes $n_k$, under Assumption 5, equivalent to Figure 7.

| Estimator | MSE ($n_k$=20) | MSE ($n_k$=60) | MSE($n_k$=100) |
|---|---|---|---|
| **Doubly Robust Estimators (AIPW)** | | | |
| Oracle AIPW ($e^*(X, Z)$) | 0.015 | 0.005 | 0.002 |
| Meta AIPW ($\hat{e}(X)$) | 10.673 | 0.346 | 0.084 |
| Fed. AIPW (Logistic MW, $\hat{e}(X, Z)$) | 0.022 | 0.004 | 0.002 |
| Fed. AIPW (Logistic MW, $\hat{e}(X)$) | 2.026 | 1.348 | 0.323 |
| Fed. AIPW (NN MW, $\hat{e}(X, Z)$) | 0.020 | 0.004 | 0.002 |
| Fed. AIPW (NN MW, $\hat{e}(X)$) | 2.043 | 1.349 | 0.315 |
| **Weighting Estimators (IPW)** | | | |
| IPW Oracle ($e^*(X, Z)$) | 0.853 | 0.443 | 0.091 |
| IPW Meta-SW ($\hat{e}(X)$) | 5.176 | 0.777 | 0.590 |
| Fed. IPW (Logistic MW, $\hat{e}(X, Z)$) | 0.427 | 0.132 | 0.043 |
| Fed. IPW (Logistic MW, $\hat{e}(X)$) | 2.001 | 1.384 | 0.327 |
| Fed. IPW (NN MW, $\hat{e}(X, Z)$) | 1.083 | 0.465 | 0.206 |
| Fed. IPW (NN MW, $\hat{e}(X)$) | 2.100 | 1.418 | 0.356 |

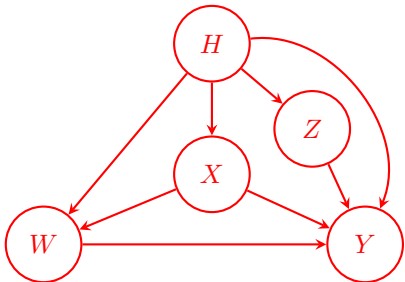

Figure 8: Graphical model for multi-site observational data where site effects are not fully explained by $Z$ (direct $H \to Y$ edge).

