# OpenReview forum: "Federated Causal Inference on Multi-Site Observational Data via Propensity Score Aggregation"
_ICLR.cc/2026/Conference — Submitted to ICLR 2026_

### Official Review · Reviewer_WHsg · 2025-10-31

**Soundness:** 4
**Presentation:** 3
**Contribution:** 2
**Rating:** 2
**Confidence:** 4

**Summary:**

This work presents a way to perform federated learning of average treatment effects. Its main novelty over existing results in the literature is to directly estimate membership weight nuisance, rather indirectly estimating it through density functions. I agree with the authors that this approach - which estimates as simple a nuisances as possible - appears to be the ideal approach.

The identifiability results and efficiency guarantees seem to follow standard arguments.

**Strengths:**

Estimating the densities used to define the nuisance is impractical: it requires tackling a harder statistical estimation problem than is necessary, and, if the densities are estimated flexibly, also requires sharing too much data across sites to classify as a "federated learning" approach. Estimating membership weights avoids these issues.

**Weaknesses:**

By Bayes' rule, $\omega_k^{\textnormal{DW}}(x)=\omega_k^{\textnormal{MW}}(x)$, so I'm unclear why different notation is being used for the two sets of weights. My understanding of your contribution is that you directly estimate the weight, rather than estimating the three components---$\rho_k,f_k,f$---used in the first representation. But the presentation obscures that.

The general idea to re-express weights in terms of source membership probabilities is commonly used in the data fusion / generalizability / transportability literatures. E.g., in slightly different problems than the one considered:

* Cole, Stephen R., and Elizabeth A. Stuart. "Generalizing evidence from randomized clinical trials to target populations: the ACTG 320 trial." American journal of epidemiology 172.1 (2010): 107-115.
* Westreich, Daniel, et al. "Transportability of trial results using inverse odds of sampling weights." American journal of epidemiology 186.8 (2017): 1010-1014.

Overall the contribution seems marginal to me. The identifiability and efficiency results are standard, so the only improvement over standard practice seems to be how one nuisance is estimated.

**Questions:**

Sorry if I'm missing something, but should Assumption 2 also condition on site ($H$)? Otherwise I don't see why Assumptions 1 and 2 together would yield $E[Y|X,W=w,H=k]=E[Y|X,W=w,H=k']$ as claimed on line 144.

---

> ### Author Response · Authors · 2025-11-21
> **Response to WHsg**
>
> We believe you may have missed the core contribution of our work, which operates in a strict federated setting—no raw data shared at any point—unlike the works you cited. The method is intentionally simple, flexible and effective: it uses off-the-shelf FL tools, works with or without modeling assumptions, remains fully federated, and is applicable to any causal estimand (not only risk difference). We restate our contributions and novelty compared to existing works, and answer your question below.
>
> ### Contributions & Positioning
> As our focus is on federated causal inference, our approach naturally borrows ideas from both federated learning and causal inference. Yet, our approach stands out for its flexibility and its simplicity of implementation in a federated context:
> 1. It allows the local propensity scores $e_k$ to take arbitrary forms and to vary freely across sites (unlike [1] and [4] who rely on strong parametric modelling of the propensity scores, potentially with known shared parameters).
> 2. It is flexible and truly satisfies the federated learning requirement that no raw data should be shared, unlike approaches based on density ratio estimation that are either highly parametric or for which no federated algorithm exist [3,4]
> 3. It does not require local overlap (unlike [1,4]), and to the best of our knowledge, is the first to provably improve overlap compared to meta-analysis.
>
> ### DW vs. MW
> It is correct that $\omega_k^{\mathrm{DW}}(x)=\omega_k^{\mathrm{MW}}(x)$ by Bayes. We use both names to highlight **estimation differences**: density-ratio estimation typically requires strong modeling assumptions, whereas membership weights $P(H\mid X)$ are **directly learnable via FL classification**, mitigating privacy issues and communication constraints.
>
> ### MW in the literature
> Prior works from transportability litterature---such as those you mention--- employ the equivalent of membership weights for single-source to single-target transport, typically in randomized trials. Our problem is different: we have multiple sites and we use MW to federate site-specific propensities into a global $e(X)$—not to transport effects to an external target. Moreover, we operate in observational settings, where identification hinges on confounding control rather than randomization, leading to distinct assumptions, estimators, and guarantees.
>
> ### Assumption 2 (clarification)
>
> Our Assumption 2 is the no-site-outcome association given covariates: $(Y(0),Y(1))\perp H\mid X$. Together with unconfoundedness, $(Y(0),Y(1))\perp W\mid X$, and consistency, we have $\mathbb{E}[Y\mid X,W=w,H=k]=\mathbb{E}[Y(w)\mid X,H=k] =\mathbb{E}[Y(w)\mid X],$
>  and thus $\mathbb{E}[Y\mid X,W=w,H=k]=\mathbb{E}[Y\mid X,W=w,H=k']$. On this matter, see our answer to reviewer LapF where we discuss relaxations of this assumption.
>
> We sincerely hope these clarifications help convey the scope and novelty of our fully federated approach. We would greatly appreciate it if you could reconsider your evaluation in light of aspects that may not have been fully apparent on first reading.
>
> [1] Xiong, R., Koenecke, A., Powell, M., Shen, Z., Vogelstein, J.T. and Athey, S., 2023. Federated causal inference in heterogeneous observational data. Statistics in Medicine,
>
> [2] Guo, T., Karimireddy, S. P., & Jordan, M. I. (2024). Collaborative heterogeneous causal inference beyond meta-analysis. arXiv:2404.15746.
>
> [3] Changchang Yin, Hong-You Chen, Wei-Lun Chao, & Ping Zhang (2025). Federated inverse probability treatment weighting for individual treatment effect estimation. arXiv:2503.04946.

---

> ### Comment · Reviewer_WHsg · 2025-11-21
>
> Thank you for your replies. I'll look through the others later, but I'm quite certain that Assumptions 1, 2, and consistency do not imply that $E[Y|X,W=w,H=k]= E[Y|X,W=w,H=k']$ for $k\not=k'$. This is essentially the same as saying that pairwise marginal independence does not imply joint independence.
>
> To make this explicit, consider the following counterexample, which is based on a classical one. Take $X$ to be trivial (e.g., $X=0$ a.s.) and let
>
> $H\\sim \\mathrm{Bern}(1/2)$,
>
> $W\\sim \\mathrm{Bern}(1/2)$,
>
> $Y(0)=Y(1)=H\\oplus W$, ($\\oplus$=XOR)
>
> $Y=WY(1) + (1-W)Y(0)$.
>
> Assumptions 1, 2, and consistency all hold. But, for $w,k\\in \\{0,1\\}$,
>
> $E[Y|X,W=w,H=k]$
>
> $= E[Y(w)|X,W=w,H=k]$ (consistency)
>
> $= w\\oplus k$. ($Y(w)=H\\oplus W$)
>
> Hence,
>
> $E[Y|X,W=w,H=0]\\not= E[Y|X,W=w,H=1]$.

---

> > ### Author Response · Authors · 2025-11-24
> >
> > Thank you for your remark. To avoid any ambiguity, we clarified that Assumption 2 concerns marginals of potential outcomes, not the joint distribution. To make this clear we updated the notations from () to {}, i.e., { $Y(0), Y(1)$ }. Additionally, we corrected an oversight in Assumption 1 to explicitly condition on $H$, i.e., $Y(w) \perp W \mid X, H$, strictly aligning the definition with our structural model in the DAG Fig. 1 which allows an arrow $H \to W$. As shown by [1] (Lemma 4.3), combining this corrected Assumption 1 with Assumption 2 implies $\{Y(w) \perp H \mid X, W\}$. This formally precludes the counterexample and guarantees the testable hypothesis holds, without changing our estimators, proofs or results.
> >
> > [1]  Dawid, A.P., 1979. Conditional independence in statistical theory. Journal of the Royal Statistical Society Series B: Statistical Methodology, 41(1), pp.1-15.

---

> > > ### Comment · Reviewer_WHsg · 2025-11-24
> > >
> > > Thank you for fixing this. When paired with Assumption 1, this correction is indeed equivalent to the original modification that I asked about in my question, since in either case the two conditions together are equivalent to the joint conditional independence $\\{Y(0),Y(1)\\}\perp (H,W)\mid X$.
> > >
> > > I'll read and respond to your other replies this week.

---

> > > > ### Comment · Reviewer_WHsg · 2025-11-24
> > > >
> > > > **DW vs. MW:** Thank you for confirming that $\\omega_k^{\mathrm{DW}}=\\omega_k^{\mathrm{MW}}$. It strikes me as strange (and nonstandard) to use two different notations to refer to the same mathematical object. I would recommend instead defining a single object, such as $\\omega_k$, to denote the object. Then, you can have two different notations for the estimators—say, $\\widehat{\\omega}_k^{\mathrm{DW}}$ and $\\widehat{\\omega}_k^{\mathrm{MW}}$—since they are indeed different.
> > > >
> > > > I also appreciate the other points you made regarding novelty. While I still feel the contribution is modest relative to the existing transportability/data fusion/generalizability literatures—which have already applied Bayes rule to notice that the membership weight parameterization may be more easily estimable—I take the author's point that the objective here is somewhat different.
> > > >
> > > > I'll raise my score to a 4.

---

> > > > > ### Author Response · Authors · 2025-11-28
> > > > >
> > > > > Thank you for raising your score.
> > > > >
> > > > > We also agree that the inconsistency in notation for the federation weights was confusing, so we have implemented in the revised version of the pdf a unified and consistent notation, defining the weights uniquely as $\omega_k$ at the population level. They now take the corresponding superscripts when estimated (i.e., $\hat\omega_k^\text{MW}$ and $\hat\omega_k^\text{DW}$).

---

### Official Review · Reviewer_LapF · 2025-11-01

**Soundness:** 3
**Presentation:** 3
**Contribution:** 2
**Rating:** 4
**Confidence:** 5

**Summary:**

This paper studies average treatment effect (ATE) estimation from decentralized multi-site observational data. The core idea is to form a global propensity score as a weighted mixture of site-specific propensity scores, with weights learned as membership probabilities via federated training; the resulting score is used in Fed-IPW and Fed-AIPW estimators. The paper shows oracle equivalence to centralized estimators, variance advantages over meta-analysis under local overlap, and improved global overlap, and provides simulations and a Traumabase data application. Empirical story is promising and exposition is clear, but the conceptual novelty beyond prior federated causal work feels incremental and several modeling choices and assumptions (especially “ignorability on sites”) should be relaxed or at least examined via sensitivity analysis procedures.

**Strengths:**

1. Practical and easily understandable by implementers. The 2-step pipeline (local PS at each site + federated membership weights) can be implemented with off-the-shelf tools; works even when some sites have poor/zero local overlap.

2. Simulations and a 14-site real dataset indicate the approach works and outperforms simple meta-analysis in variance and robustness.

**Weaknesses:**

1. Outcome modeling and AIPW components mirror prior work (e.g., Khellaf et al.) and the membership-weight mixture of local PS, while sensible, feels like a straightforward extension; the paper needs crisper positioning against center-effect / concept shift and transportability literatures and federated PS aggregation (e.g., parameter or consensus / voting approaches)

2. Estimating local PS's P(H=k | X=x) can be problematic when many sites have this probability close to 0; common in multi-institution data. The paper acknowledges density-ratio issues but underplays analogous small-probability problems and potential co-training or regularization strategies.

3. Real-data models (logistic PS, FedAvg logistic outcomes; a 1-hidden-layer, 128-unit NN for MW) seem overly simplistic

4. No reference to code or proofs; no appendix.

**Questions:**

* How would the DAG change if you have center-level covariates, and how would nuisance models be estimated potentially differently as a result? I believe that you could have weaker assumptions leveraging center-level covariates that could then allow for a direct arrow from H to Y. Alternatively, would it be sufficient to assume (Y(0), Y(1) \perp W | (H,X) or a mixed structure (e.g., random intercepts) and show when your Fed-(A)IPW remains consistent? What sensitivity analysis do you recommend if Assumption 2 fails?

* What happens when some local PS models are misspecified and the MW classifier can also be misspecified? Theoretical results for rate-double-robust style guarantees (e.g., products of estimation errors) for Fed-AIPW in the federated setting would be helpful.

* Instead of training K independent PS models, can you co-train subsets of sites that are similar (multi-task or clustered FL) and mix only across clusters? How would that change variance and bias?

* Any theory comparing the asymptotic efficiency of Fed-(A)IPW to (i) the best single-site estimator and (ii) pooled oracle, beyond the variance comparisons to meta-analysis?

---

> ### Author Response · Authors · 2025-11-21
> **Response to Reviewer LapF (1)**
>
> We thank the reviewer for the careful assessment of our method’s soundness, clarity, and theoretical guarantees. We address their remarks and questions below.
>
> ### W1. Positioning:
> * Our approach stands out for its flexibility and its simplicity of implementation in a federated context:
> 1. It allows the local propensity scores $e_k$ to take arbitrary forms and to vary freely across sites (unlike [1] and [4] who rely on strong parametric modelling of the propensity scores, potentially with known shared parameters).
> 2. It is flexible and truly satisfies the federated learning requirement that no raw data should be shared, unlike approaches inherited from transportability domains, which are based on density ratio estimation that are either highly parametric or for which no federated algorithm exist [2,3,4].
> 3. It does not require local overlap (unlike [1,4]), and to the best of our knowledge, is the first to provably improve overlap compared to meta-analysis.
> * Specifically, our MW-based mixture of heterogeneous local propensity scores learned with standard FL requires no modeled shared structure and no local overlap, unlike (i) meta-analysis (which fails without local overlap) and (ii) parameter/consensus propensity score approaches (which assume common parametric forms or voting across “compatible” sites).
> * Finally, as we discuss in the Questions section below, our approach can be adapted to handle site effects when they exist.
>
> ### W2. Small $P(H=k|X)$:
> * We evaluate membership $P(H\mid X)$ at observed covariates $x$ from the site mixture, so for each $x$ there is typically one site with a high posterior $P(H=k\mid X=x)$ (the site that generated $x$) in moderate dimensions. Our NN with temperature scaling captures near-zero posteriors well, and sites with tiny $P(H=k\mid x)$ naturally receive negligible weight in
> $\hat e(x)=\sum_{k=1}^K P(H{=}k\mid x)\hat e_k(x),$
> where the MW are convex weights (positive and summing to 1).
> * In high-dimensional settings with many sites, posteriors can become uniformly small due to sample-complexity limits. In such cases, regularization is essential, and MW provide a more flexible way to introduce calibration, shrinkage, or smoothing across sites than  density-ratio weights.
>
> ### W3. Real-data:
> * "Real-data models seem overly simplistic": we learn MW with federated NNs (also feasible with federated RFs). Several sites have small $n_k$, so local probability forests (grf package) yield unstable estimates of $e_k$ for such sites. We therefore estimated the $e_k$ with simple logistics, and a compact 2-layer NN for MW. Despite this simplicity, Fed-NN MW (A)IPW match a centralized fully nonparametric benchmark (causal forest AIPW estimated on the pooled dataset, which would be infeasible in practice in a federated setting). Per your request, we re-ran the study using RF-based MW and probability-forest $e_k$'s on sites with enough data, and logistic for smaller sites. The updated results (Fig. 4) show the same trend as those of our previous experiment, comforting us in the flexilibity of our approach.
>
> ### W4. Code and proofs
> * We have included these in the supplementary .zip since the submission. You will find there the proofs, additional experiments, and pseudo-code for learning MW in federated settings.

---

> > ### Author Response · Authors · 2025-11-21
> > **Response to Reviewer LapF (2)**
> >
> > ### Q1. Extension to site effects:
> >
> > First, **Assumption 2 is testable**. One can fit a federated regression $Y \sim X + H$ and test whether the site coefficients are zero conditional on $X$. In FL, this can be done via one-shot sufficient-statistics sharing—each site transmits $X^\top X$, $X^\top Y$ -- then a joint test (F/Wald) on the $H$ block provides a $p$-value. We therefore recommend adapting the estimation strategy depending on the outcome of this test.
> >
> > Second, **our approach can be adapted to handle site-effects when site-level covariates $Z$ are available**. In this case, we add $H \to Z \to Y$ to the DAG in Figure 1. Then, a weaker and testable replacement for Assumption 2 is
> > $$(Y(1),Y(0))\perp H\mid (X,Z),$$
> > i.e., $(X,Z)$ is a sufficient adjustment set. Under this relaxed assumption, we only need to change how we form the global propensity score: $$e(X,Z)=\sum_{k=1}^K P(H{=}k\mid X,Z)e_k(X).$$
> > In other words, MW are fit using $(X,Z)$ to predict $P(H\mid X,Z)$, while the local propensities $e_k$ stay functions of $X$ (since within a site, $Z$ is constant). For AIPW, outcome models are fit on $(X,Z)$ in the usual federated fashion, yielding $\mu_w(X,Z)$.
> >
> > To showcase the adaptivity of our method to site effects under this weaker assumption, we simulated $K=30$ sites, each holding site-level covariates $Z\in\mathbb{R}^3$ drawn from a site-specific Gaussian (modeling site effects) and individual-level covariates $X\in\mathbb{R}^3$ (differing across sites to simulate covariate shift). Treatments are assigned with heterogeneous $e_k$ functions. Please find details on the parameters at the end of this post. To estimate membership $P(H=k\mid \cdot)$, we fit either a multinomial logit or a 2-layer NN using features $(X)$ or $(X,Z)$. For each site, we fit local propensity models $\hat e_k(x)$ by logistic regression on $(X)$, and local outcome models $\hat\mu_{w,k}(\cdot)$ by OLS on $(X)$. Membership weights (MW) aggregation forms a global propensity $\hat e_{\mathrm{MW}}(x,z)=\sum_k \hat P(H=k\mid x,z)\hat e_k(x)$; for Fed-AIPW we also use a federated OLS $\hat\mu_w(x,z)$. For meta-analysis, we compute site-specific estimates $\hat\tau_k$ with locally estimated nuisance functions $e_k(\cdot)$ and average them with sample-size weights $n_k/n$. The results are as follows:
> >
> > | Estimator                  | MSE (n_k=20) | MSE (n_k=60) | MSE (n_k=100) |
> > |:-------------------------- | ------------:| ------------:| -------------:|
> > | Fed-AIPW MW(NN)[X,Z]       |         0.02 |        0.004 |         0.002 |
> > | Fed-AIPW MW(NN)[X]         |        2.043 |        1.349 |         0.315 |
> > | Fed-AIPW MW(logistic)[X,Z] |        0.022 |        0.004 |         0.002 |
> > | Fed-AIPW MW(logistic)[X]   |        2.026 |        1.348 |         0.323 |
> > | AIPW Meta-SW[X]            |       10.673 |        0.346 |         0.084 |
> > | AIPW Oracle[e*]            |        0.015 |        0.005 |         0.002 |
> > | Fed-IPW MW(NN)[X,Z]        |        1.083 |        0.465 |         0.206 |
> > | Fed-IPW MW(NN)[X]          |          2.1 |        1.418 |         0.356 |
> > | Fed-IPW MW(logistic)[X,Z]  |        0.427 |        0.132 |         0.043 |
> > | Fed-IPW MW(logistic)[X]    |        2.001 |        1.384 |         0.327 |
> > | IPW Meta-SW[X]             |        5.176 |        0.777 |         0.590 |
> > | IPW Oracle[e*]             |        0.853 |        0.443 |         0.091 |
> >
> > * Adjusting via $\hat e(x,z)=\sum_{k=1}^K \hat P(H=k\mid x,z)\hat e_k(x)$ uses $Z$ as a proxy for latent $H$, so $(X,Z)$-based membership removes confounding from both covariate shift ($H\to X$) and site effects ($H\to Z\to Y)$ without putting $Z$ into the local propensities $\hat e_k(X)$. Under the testable condition $(Y(1),Y(0))\perp H\mid(X,Z)$ this yields unbiasedness of $(X,Z)$-adjusted estimators, whereas those who condition only on $X$ are biased.
> > * Fed-AIPW MW with $(X,Z)$ are effectively unbiased and attain very low MSE because a well-specified $\hat\mu_w(X,Z)$ absorbs residual imbalance even when $\hat e$ is noisy.
> > * Meta estimators suffer with small $n_k$ because they rely on noisy per-site nuisance fits ($\hat e_k(x),\hat\mu_{w,k}(x)$). Federated MW estimators borrow strength across all sites (learned on $n=\sum_k n_k$ samples), yielding much lower MSE even when $n_k$ is small.
> > * Fed-IPW with $(X,Z)$ shows declining MSE as $n_k$ grows. Multinomial logit is sample-efficient when correctly specified; neural networks need more data but avoid parametric assumptions on $P(H\mid X,Z)$ and perform reasonably well in small $n_k$ here too.
> >
> > Finally, if a direct $H \to Y$ remains (or if $Z$ variables are not available), we can assume $(Y(1),Y(0)) \perp W \mid (X,H)$ and learn $e(X,H)$ and $\mu_w(X,H)$ federatively with parametric models. However, in this setting, we should not expect much gain from federated learning compared to meta-analysis, which naturally stratifies on $H$.

---

> > > ### Author Response · Authors · 2025-11-21
> > > **Response to Reviewer LapF (3)**
> > >
> > > ### Other questions
> > >
> > > * **Misspecifications:** In Appendix E.2, we show our method can be fully model-agnostic, avoiding misspecification issues; in Appendix E.3, we show robustness to local inconsistencies in propensity scores as $K$ grows.
> > >
> > > * **Clustered-FL extension**: we can implement a clustered-FL version by defining $C$ clusters of similar propensity score sites and learning cluster-level $e_c(X)$ either federatively within cluster $c$, or via $e_c(X)=\sum_{k\in c} P(H{=}k\mid X,C),e_k(X)$. Then, we form $e(X)=\sum_{c=1}^C P(C|X)e_c(X)$ where cluster-membership weights are learned with our method. This should lower the variance the more clusters are heterogeneous in $e_c$'s.
> > >
> > > We thank the reviewer for the thoughtful and constructive feedback. The remarks on site effects strengthen our contribution as it allows to naturally extend our federated approach to a broader setting. We hope the breadth of experiments, the real-data analysis, and the use of testable assumptions convey the practical relevance and robustness of our method. We would sincerely appreciate a positive reassessment.
> > >
> > > -----
> > >
> > > [1] Xiong, R., Koenecke, A., Powell, M., Shen, Z., Vogelstein, J.T. and Athey, S., 2023. Federated causal inference in heterogeneous observational data. Statistics in Medicine,
> > >
> > > [2] Guo, T., Karimireddy, S. P., & Jordan, M. I. (2024). Collaborative heterogeneous causal inference beyond meta-analysis. arXiv:2404.15746.
> > >
> > > [3] Changchang Yin, Hong-You Chen, Wei-Lun Chao, & Ping Zhang (2025). Federated inverse probability treatment weighting for individual treatment effect estimation. arXiv:2503.04946.
> > >
> > > [4] Larry Han, Jue Hou, Kelly Cho, Rui Duan, and Tianxi Cai. Federated adaptive causal estimation of target treatment effects. Journal of the American Statistical Association
> > >
> > > Simulation setup: we simulate $K=30$ sites, each holding $n_k=\{30, 90, 150\}$ observations. We generate first the covariates $X\in\mathbb{R}^3\sim \mathcal{N}(0,I_3)$ drawn i.i.d., before allocating them a site $H\sim Logit(\theta^\top X)$ to simulate covariate shift. We also draw site-covariates (site effects) $Z\in\mathbb{R}^3$ from a site-specific Gaussian: for each site $k$, a center $\mu_k\sim\mathcal{N}([1,1,1], I_3)$, then $Z\mid H=k\sim\mathcal N(\mu_k, I_3)$. Treatment is assigned by a site-specific logistic model, outcomes are linear: $Y=X^\top\beta+Z^\top\delta+\tau W+\varepsilon$, with $\varepsilon\sim\mathcal N(0,1)$, and we set $\tau=10$. We use $\beta=\delta=[-2,-1.33,-.66]$. We compute IPW and AIPW as $\hat\tau_{\mathrm{IPW}}=\frac{1}{n}\sum_i\frac{W_i Y_i}{\hat e(X_i)}-\frac{1}{n}\sum_i\frac{(1-W_i)Y_i}{1-\hat e(X_i)}$ and $\hat\tau_{\mathrm{AIPW}}=\frac{1}{N}\sum_i\big[\frac{W_i(Y_i-\hat\mu_1)}{\hat e(X_i)}-\frac{(1-W_i)(Y_i-\hat\mu_0)}{1-\hat e(X_i)}+(\hat\mu_1-\hat\mu_0)\big]$.

---

### Official Review · Reviewer_ue8S · 2025-11-04

**Soundness:** 3
**Presentation:** 3
**Contribution:** 3
**Rating:** 6
**Confidence:** 5

**Summary:**

This paper introduces a federated approach for estimating the Average Treatment Effect (ATE) from decentralized observational data without sharing individual-level records. The authors propose Federated IPW (Fed-IPW) and Federated AIPW (Fed-AIPW) estimators that aggregate locally estimated propensity scores using Membership Weights (MW)—the probability of site membership conditional on covariates. Compared to density ratio weighting (DW) methods from transportability literature, MW can be flexibly estimated via standard federated learning algorithms using either parametric or nonparametric models, thereby improving overlap and robustness.
The paper provides theoretical guarantees showing that the proposed estimators attain the same efficiency as centralized ones, outperform meta-analysis under weaker overlap assumptions, and empirically validate the method on both synthetic and real datasets (Traumabase).

**Strengths:**

1. Federated causal inference is an important emerging area where privacy-preserving data analysis meets causal estimation. The paper correctly identifies limitations of current meta-analysis and density-ratio-based approaches and proposes a plausible solution.

2. The proposed Membership Weight aggregation is mathematically well-defined, with clear derivations showing variance reduction (Theorems 3–5) and improved overlap. The proofs appear sound and the arguments are intuitive (e.g., the toy example on page 6 clearly demonstrates the “global overlap improvement” effect).

3. The simulations across varying overlap regimes (no, poor, good local overlap) convincingly illustrate that Fed-IPW and Fed-AIPW remain unbiased and stable even when meta-analysis fails (Figures 2–4).

**Weaknesses:**

1. Lack of in-depth discussion between prior work. The contribution mainly lies in the MW-based aggregation strategy. While elegant, it largely builds on existing AIPW/IPW machinery and the federated estimation literature. I personally feel that the method proposed in the paper is very closely related to prior work by Guo et al. (2024). Although they considered estimating the ATE w.r.t. the target distribution. When taking the target distribution as the entire population, the two proposed estimators look similar.

2. Empirical analysis could be deeper:

- The synthetic experiments mainly vary overlap conditions but do not explore heterogeneity in nuisance models (e.g., nonlinear treatment mechanisms or site-specific confounding structures).

- The real-data example (Traumabase) is informative but somewhat limited: the analysis is primarily AIPW-based with simple logistic regressions. It would be useful to examine nonparametric MW models more thoroughly or report sensitivity to misspecification.

3. Assumption 2 (ignorability on sites) seems strong: In realistic federated healthcare scenarios, site membership may directly affect outcomes (e.g., hospital quality or local protocols). There exists literature in Meta-analysis that uses L1-regulizer style estimator to deal with this issue. It is worthwhile to incorporating that in the setting.

**Questions:**

Can the authors compare their estimators and the estimators in Guo et al. (2024), when their target site becomes the super-population defined by the K participating sites?

---

> ### Author Response · Authors · 2025-11-21
> **Response to Reviewer ue8S (1)**
>
> We are glad that you appreciated our work, its soundness and its presentation. We are especially grateful for your careful recognition of the article's strengths, including your thorough examination of the proofs. We address below your remarks and questions in order.
>
> ### 1. Positioning
> * **General position wrt related work.** As our focus is on federated causal inference, our approach naturally borrows ideas from both federated learning and causal inference. Yet, our approach stands out for its flexibility and its simplicity of implementation in a federated context:
> 1. It allows the local propensity scores $e_k$ to take arbitrary forms and to vary freely across sites (unlike [1] and [4] who rely on strong parametric modelling of the propensity scores, potentially with known shared parameters).
> 2. It is flexible and truly satisfies the federated learning requirement that no raw data should be shared, unlike approaches based on density ratio estimation that are either highly parametric or for which no federated algorithm exist [2,3,4].
> 3. It does not require local overlap (unlike [1,4]), and to the best of our knowledge, is the first to provably improve overlap compared to meta-analysis.
> * **Comparison to Guo et al. applied to our setting.** The setting considered in [2] does include the special case where the target is the super-population of the $K$ sites, but even then, the core differences with our approach remain. Their method requires access to the target's covariates through an external public dataset. It also requires estimating a separate density ratio for each site and each treatment arm, yielding $2K$ model fits to learn $e$. These ratios are learned directly, not via density estimation, using either parametric models with strong assumptions (the "exponential tilt" form, essentially a logistic model fit by moment-matching, sensitive to misspecification and untestable in this setup) or nonparametric methods (such as KNN) that cannot be run without sharing raw covariates—incompatible with the constraints of FL. In contrast, our approach fits only one model per site, learns MW via standard FL classifiers, and never shares covariate information.
>
> ### 2. Experiments:
> * **"The synthetic experiments do not explore heterogeneity in nuisance models"**: Propensities $e_k$ are site-specific and can take any form—our simulations draw them from distinct logistic models with different parameters (Appendix C)—and they can be estimated with flexible learners (e.g., random forests; see Appendix E.3). Under Assumption 2 (no site effect), outcomes $\mu_1,\mu_0$ are shared across sites; however, the framework readily accommodates heterogeneous $\mu_{1,k}, \mu_{0,k}$ when Assumption 2 is relaxed (see the discussion on relaxing Assumption 2 below).
>
> * **On real-data with simple logistic regressions, and on using nonparametric models for MW**:
>     * **Traumabase data.** Several sites have small $n_k$, so complex models like probability forests (grf package) yield unstable estimates of $e_k$ for such sites. We therefore estimated the $e_k$ with simple logistics, and a compact 2-layer NN for MW. Despite this simplicity, Fed-NN MW (A)IPW match a centralized fully nonparametric benchmark (causal forest AIPW estimated on the pooled dataset, which would be infeasible in practice in a federated setting). We also include a random-forest MW estimator as a complementary, nonparametric benchmark—run centrally (federated RF is outside our scope). The updated results (Fig. 4) mirror our original findings: Fed-MW AIPW remains aligned with the centralized nonparametric benchmark, while performance is robust to the choice of local PS learner—underscoring the flexibility of our approach.
>     * **Additional synthetic experiments with nonparametric MW models and robustness to misspecification.** In additional simulations (Appendix E.2), we estimate the ATE via model-agnostic procedures: local $e_k$ are estimated with generalized random forests and MW with a federated NN, avoiding misspecification issues while respecting FL constraints. In Appendix E.3, we further show robustness to local propensity score inconsistencies: even when we intentionally misspecify site 1’s propensity score as $\hat e^1(x)=0.1$, the ATE error shrinks toward zero as $K$ grows (with fixed $n_k$).

---

> > ### Author Response · Authors · 2025-11-21
> > **Response to Reviewer ue8S (2)**
> >
> > ### 3. Assumption 2:
> > * Indeed, Assumption 2 implies that $H$ influences $Y$ only through $X$, and has thus no direct effect on $Y$ (which we refer to as "no-site-effect") can sometimes be violated (e.g., quality or budget differences across sites); however it is testable. One can indeed run a federated regression $Y \sim X + H$ and test whether site effects vanish conditional on $X$. This can be done in a one shot procedure via sharing sufficient statistics $X^\top X$, $X^\top Y$, and terms for $H$; a joint Wald/F-test provides a $p$-value. In moderate dimensions, this is reliable; in higher dimensions, a ridge-regularized federated OLS test is also feasible. We therefore recommend adapting the estimation strategy depending on the outcome of this test.
> > * Please refer to the response to Reviewer LapF where we show that our method can be adapted to the presence of site effects when site-covariates are available. We provide simulations where we allow the outcome response functions to be heterogeneous across sites.
> > * We would be grateful if you could share the reference for L1 regularizers in meta analysis for site effect penalization, although this may be more closely related to random-effects meta-analysis—which, to our knowledge, has not yet been developed in the FL context.
> >
> > We hope these clarifications help clarify our contributions and allow you to view them even more favorably. Thank you again for the time and care you devoted to this review.
> >
> > [1] Xiong, R., Koenecke, A., Powell, M., Shen, Z., Vogelstein, J.T. and Athey, S., 2023. Federated causal inference in heterogeneous observational data. Statistics in Medicine,
> >
> > [2] Guo, T., Karimireddy, S. P., and Jordan, M. I. (2024). Collaborative heterogeneous causal inference beyond meta-analysis. arXiv preprint arXiv:2404.15746.
> >
> > [3] Changchang Yin, Hong-You Chen, Wei-Lun Chao, and Ping Zhang. Federated inverse probability treatment weighting for individual treatment effect estimation. arXiv preprint arXiv:2503.04946, 2025.
> >
> > [4] Larry Han, Jue Hou, Kelly Cho, Rui Duan, and Tianxi Cai. Federated adaptive causal estimation of target treatment effects. Journal of the American Statistical Association

---

### Author Response · Authors · 2025-11-28
**Global response**

We sincerely thank the reviewers for their constructive feedback, which has strengthened our contributions. We have uploaded a revised manuscript with changes highlighted in red to address the concerns raised. Specifically, we have made the following updates:

* **Site Effects (Reviewers LapF and ue8S):** We added a comprehensive section in the appendix detailing how our framework can incorporate site-level covariates to explicitly model site effects. We also updated Section 2.2 to explicitly reference this extension when defining Assumption 2.
* **Real Data Analysis (Reviewer ue8S):** We updated the real data analysis to include Random Forest Membership Weights, demonstrating that non-parametric approaches yield results equivalent to the simpler models (e.g., 2-layer Neural Networks) previously used. Additionally, we refined the estimation of local propensity scores ($e_k$) to use probability forests (GRF) for sites with sufficient data, while retaining logistic regression for smaller sites to ensure stability.
* **Notation and Definitions (Reviewer WHsg):** We implemented a unified notation for federation weights (defined uniquely as $\omega_k$ at the population level, with superscripts when estimated) and corrected the definition of Assumption 1 to explicitly condition on the site.

---

### Meta-Review · Area_Chair_98ah · 2026-01-05

**Summary:**

All the reviews were of exceptionally high quality and the authors made concerted efforts to address and incorporate them. However, the reviewer consensus is that this work has incremental conceptual novelty - LapF and WHsg note this explicitly, and ue8S questions its improvement over (Guo et al. '24).

Some other reviewer concerns that were unaddressed in rebuttal:
- Please add a more thorough discussion related to (Guo et al. '24), especially as the author's admit the current setting is a special case of theirs.
- Add the discussion on Assumption 2 and center-level covariates to the final paper
- Add asymptotic efficiency analysis of Fed-(A)IPW and compare to known results from meta-analysis.
- Please carefully go through the comments below and add discussions on anything currently marked as partially addressed or not addressed. Also make sure that the discussions below are incorporated into the paper.

**Reviewer Concerns:**

- Reviewer ue8S
  - Comparison to Guo et al. 2024. Authors clarified that their method fits only one model per site and never shares raw covariate information, unlike prior work that requires external public datasets or non-parametric methods incompatible with FL. partially addressed. A direct comparision of the theoretical guarantees of the two methods was not provided.
  - Limitations of empirical analysis in non-parametric models. The revision added Random Forest Membership Weights and Probability Forests, demonstrating that the method also works with more complex non-parametric learners. addressed.
  - Strong assumption of site-level ignorability. Authors added an appendix section discussing this. addressed.
  - Direct estimator comparison to super-population targets. The response highlighted that their approach is simpler to implement in FL as it uses standard classifiers rather than complex, data-sharing-dependent density ratios. addressed.


- Reviewer LapF
  - Incremental conceptual novelty. Authors emphasized being the first fully federated method to provably improve overlap over meta-analysis without requiring local overlap at every site. partially addressed. See eeviewer WHsg's comment that they "still feel the contribution is modest...".
  - Instability from near-zero site membership probabilities. They argued that membership weights naturally handle these cases by assigning negligible weights and offer better calibration than density-ratio methods. addressed.
  - Simplistic real-data modeling. The Traumabase analysis was updated with Random Forests, showing results remain consistent across more complex models. addressed.
  - Impact of center-level covariates. A mathematical extension was provided showing that site-level covariates can act as proxies for site effects, effectively removing bias in new simulations. addressed.
  - Risks of model misspecification. New appendix sections demonstrate the method's robustness to local propensity score inconsistencies and its model-agnostic nature. partially addressed (no theory only simulations).
  - Clustered FL feasibility. The response clarifies indeed this is possible. addressed.
  - Asymptotic efficiency of Fed-(A)IPW. seems to have been overlooked. not addressed.


- Reviewer WHsg
  - Confusing notation for DW and MW. Authors implemented a unified notation ($\omega_k$) for population-level weights in the revised manuscript to improve clarity.
  - Modest contribution to transportability literature. They clarified that applying these concepts to multi-site federated observational data with strict privacy constraints presents a distinct, novel challenge.
  - Conditional independence oversight in Assumption 2. The authors corrected Assumption 1 to explicitly condition on the site, which mathematically resolved the "XOR counterexample" raised.

**Reviewer Scores:**

- Reviewer ue8S would likely have retained their **6** since their main concern was comparision with (Guo et al. 2024)
- Reviewer LapF may have at best increased to **6**. Two of their questions not addressed, but the authors made concerted efforts to address the rest and improve the paper.
- Reviewer WHsg explicitly states they raise their score to **4**

---

### Decision · Program_Chairs · 2026-01-26

Reject